# Flows for simultaneous manifold learning and density estimation

**Johann Brehmer and Kyle Cranmer**
New York University
johann.brehmer@nyu.edu, kyle.cranmer@nyu.edu

## Abstract

We introduce manifold-learning flows ($\mathcal{M}$-flows), a new class of generative models that simultaneously learn the data manifold as well as a tractable probability density on that manifold. Combining aspects of normalizing flows, GANs, autoencoders, and energy-based models, they have the potential to represent datasets with a manifold structure more faithfully and provide handles on dimensionality reduction, denoising, and out-of-distribution detection. We argue why such models should not be trained by maximum likelihood alone and present a new training algorithm that separates manifold and density updates. In a range of experiments we demonstrate how $\mathcal{M}$-flows learn the data manifold and allow for better inference than standard flows in the ambient data space.

## 1   Introduction

Inferring a probability distribution from example data is a common problem that is increasingly tackled with deep generative models. Both generative adversarial networks (GANs) [1] and variational autoencoders (VAEs) [2] are based on a lower-dimensional latent space and a learnable mapping to the data space, in essence describing a lower-dimensional data manifold embedded in the ambient data space. While they allow for efficient sampling, their probability density is intractable, limiting their usefulness for inference tasks. On the other hand, normalizing flows [3–6] are based on a latent space with

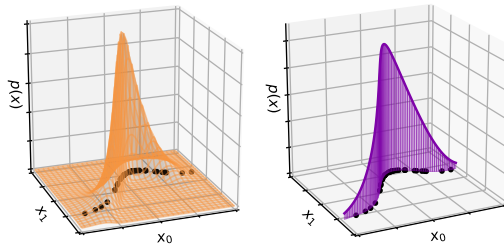

Figure 1: Sketch of how a standard flow in the ambient data space (left) and an $\mathcal{M}$-flow (right) model data on a manifold.

the same dimensionality as the data space and a diffeomorphism; their tractable density permeates the full data space and is not restricted to a lower-dimensional surface.

The flow approach may be unsuited to data that do not populate the full ambient data space they natively reside in, but are restricted to a lower-dimensional manifold [7]. Normalizing flows are by construction not able to represent such a structure exactly, instead they learn a smeared-out version with support off the data manifold. We illustrate this in the left panel of Fig. 1. In addition, the requirement of latent spaces with the same dimension as the data space increases the memory footprint and computational cost of the model. While flows have been generalized from Euclidean spaces to Riemannian manifolds [8], this approach has so far been limited to the case where the chart for the manifold is prescribed.

We introduce *manifold-learning flows* ($\mathcal{M}$-flows): normalizing flows based on an injective, invertible map from a lower-dimensional latent space to the data space. $\mathcal{M}$-flows simultaneously learn the shape of the data manifold, provide a tractable bijective chart, and learn a probability density over the

manifold, as sketched in the right panel of Fig. 1.

We discuss how this approach marries aspects of normalizing flows, GANs, autoencoders, and energy-based models [9–11]. Compared to standard flow-based generative models, $\mathcal{M}$-flows may more accurately approximate the true data distribution, avoiding probability mass off the data manifold. In addition, the model architecture naturally allows one to model a conditional density that lives on a fixed manifold. This should improve data efficiency in such situations as it is ingrained in the architecture and does not need to be learned. The lower-dimensional latent space may also reduce the complexity of the model, and the ability to project onto the data manifold provides dimensionality reduction, denoising, and out-of-distribution detection capabilities.

In this paper we summarize the main conceptual ideas and findings. We will frequently relegate in-depth discussions, additional results, and implementation details to the supplementary material, which contains a substantially extended version of this paper. The code used in our study is available at `http://github.com/johannbrehmer/manifold-flow`.

## 2    Generative models and the data manifold

Consider a data-generating process that draws samples $x \in \mathcal{M}^* \subset X = \mathbb{R}^d$ according to $x \sim p^*(x)$, where $\mathcal{M}^*$ is a $n$-dimensional Riemannian manifold embedded in the $d$-dimensional data space $X$ and $n < d$. We consider the two problems of estimating the density $p^*(x)$ as well as the manifold $\mathcal{M}^*$ given some training samples $\{x_i\} \sim p^*(x)$.

We will first review how existing classes of generative models address these problems, before introducing the new manifold-learning flows ($\mathcal{M}$-flows). For simplicity, we treat the manifold as topologically equivalent to $\mathbb{R}^n$ and assume that its dimensionality $n$ is known. To facilitate a straightforward comparison, we will describe all generative models in terms of two vectors of latent variables $u \in U$ and $v \in V$, where $U = \mathbb{R}^n$ is the latent space that maps to the learned manifold $\mathcal{M}$, i.e. the coordinates of the manifold. $V = \mathbb{R}^{d-n}$ parameterizes any remaining latent variables, representing the directions "off the manifold". We summarize the approaches in Fig. 2 and Tbl. 1, and discuss many algorithms in more detail in Sec. 2 of the supplementary material.

**Ambient flow (AF).** A standard Euclidean normalizing flow [6] in the ambient data space is a diffeomorphism $f : U \times V \mapsto X$ together with a tractable base density $p_{uv}(u, v)$. The density in $X$ is then given by $p_x(x) = p_{uv}(f^{-1}(x)) \left| \det J_f(f^{-1}(x)) \right|^{-1}$, where $J_f$ is the Jacobian of $f$, a $d \times d$ matrix. In the generative mode, flows sample $u$ and $v$ from their base densities and apply the transformation $x = f(u, v)$, yielding $x \sim p_x(x)$. There is no difference between $u$ and $v$, and the model has no notion of a data manifold.

**Flow on a prescribed manifold (FOM).** When a chart $g^* : U \mapsto \mathcal{M}^* \subset X$ for the data manifold is known a priori, one can construct a flow on this manifold [8, 12–14]. The density is only defined over $\mathcal{M}^*$ and given by $p_{\mathcal{M}^*}(x) = p_u(g^{*-1}(x)) \left| \det[J_g^T(g^{*-1}(x)) J_g(g^{*-1}(x))] \right|^{-\frac{1}{2}}$, where $J_g$ is the Jacobian of $g^*$, an $n \times d$ matrix. The density $p_u(u)$ is modeled with a standard normalizing flow in $n$ dimensions, i.e. with a learnable diffeomorphism $h$ that maps to another set of latent variables $\tilde{u}$ and a corresponding base density $p_{\tilde{u}}(\tilde{u})$.

**Generative adversarial networks (GANs)** map an $n$-dimensional latent space to the data space

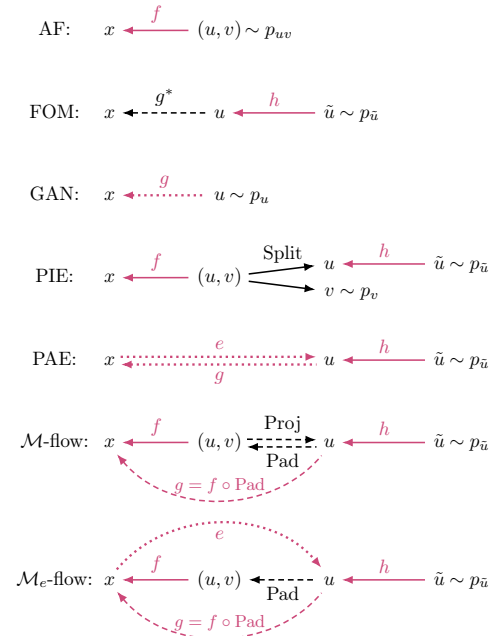

Figure 2: Schematic relation between data $x$ and latent variables $u, v, \tilde{u}$ in different models. Red arrows show learnable transformations, black arrows prescribed ones. Solid lines denote bijections, dashed lines injections, dotted lines unrestricted transformations.

through a learnable function $g : U \mapsto \mathcal{M} \subset X$. $g$ is neither restricted to be invertible nor injective, and there can be multiple $u$ that correspond to the same data point $x$. Strictly speaking, $g$ is not a chart and the image of this transformation not necessarily a Riemannian manifold, though this distinction is not our focus and we will simply call this subset a manifold. While the lack of restrictions on $g$ increases the expressivity of the neural network, it also makes the model density intractable. **Variational autoencoders (VAEs)** similarly link a lower-dimensional latent space to the data space. They can also often be associated with a learned data manifold, for instance via the mean of a Gaussian decoder; subtleties in this relation are discussed in the supplementary material.

**Pseudo-invertible encoder (PIE).** One way to give ambient flows a notion of a (learnable) data manifold is to treat some of the latent variables differently from others and rely on the training to align one class of latent variables with the manifold coordinates. The PIE model [15] splits the latent variables of an ambient flow into two vectors $u$ and $v$ with different base densities. The distribution of $u$, designated to represent the coordinates on the manifold, is modeled with an $n$-dimensional Euclidean flow, i. e. a transformation $h$ that maps it to

| Model | Manifold | Chart | Tractable density | Restr. to $\mathcal{M}$ |
|---|---|---|---|---|
| AF | no manifold | ✗ | ✓ | ✗ |
| FOM | prescribed | ✓ | ✓ | ✓ |
| GAN | learned | ✗ | ✗ | ✓ |
| VAE | learned | ✗ | only ELBO | (✗) |
| PIE | learned | ✓ | ✓ | (✗) |
| PAE | learned | ✗ | ✗ | ✓ |
| $\mathcal{M}$-flow | learned | ✓ | ✓ | ✓ |
| $\mathcal{M}_e$-flow | learned | ✓ | ✓ | ✓ |

Table 1: Comparison of generative models. The last column classifies models by whether their density is restricted to the manifold; parantheses (✗) indicate an alternative sampling procedure that can generate data restricted to the manifold, but which does not correspond to the model density.

another latent variable $\tilde{u}$ associated with some base density. For $v$, which should learn the off-the-manifold directions in latent space, it uses a base density $p_v(v)$ that sharply peaks around 0, for instance a Gaussian with a small variance $\varepsilon^2 \ll \mathrm{Var}[u]$.

For sufficiently flexible transformations, this architecture has the same expressivity as an ambient flow, independently of the orientation of the latent space. In particular, a single scaling layer can learn to absorb the difference in base densities, allowing the flow to squeeze any region of data space into the narrow base density $p_v(v)$. From that perspective it does not seem like PIE is actually a different model than AF. Yet somehow in practice learning dynamics and the inductive bias of the model seem to couple in a way that favor an alignment of the level set $v = 0$ with the data manifold.

The model density $p_x(x)$ has the same form as for AFs and generally has support over the full data space $X$, extending beyond the manifold. While one could sample from this density, the authors of Ref. [15] instead define a generative mode that samples data only from the learned manifold by sampling $u \sim p_u(u)$ and applying $x = f(u, 0)$, i. e. fixing the off-the-manifold latents to $v = 0$. Note, however, that the density defined by this sampling procedure is *not* the same as the tractable density $p_x(x)$ (and not even proportional to $p_x(x)$ for $x \in \mathcal{M}$). We discuss this inconsistency in depth in Sec. 2.C of the supplementary material.

In a conditional version of the original PIE model, both the shape of the manifold as well as the density on it generally depend on the variables $\theta$ being conditioned on. We introduce a new conditional PIE version in which $g$ (and thus the manifold) is independent of $\theta$, while $h(\tilde{u}|\theta)$ and therefore the density are conditional on $\theta$. Training such a model by maximum likelihood leads to a stronger incentive to align the variables $u$ with the manifold coordinates, since in any other alignment the model cannot model the dependence on $\theta$.

**Manifold-learning flow ($\mathcal{M}$-flow).** As the main new model presented in this paper, $\mathcal{M}$-flows combine the learnable manifold aspect of GANs with the tractable density of FOMs without introducing inconsistencies between generative mode and the tractable likelihood. We begin by modeling the relation between the latent space and data space with a diffeomorphism $f : U \times V \mapsto X$, just as for an ambient flow or PIE. We define the model manifold $\mathcal{M}$ through the level set

$$g : U \mapsto \mathcal{M} \subset X \quad \text{with} \quad u \to g(u) = f(u, 0) \,. \tag{1}$$

In practice, we implement this transformation as a zero padding followed by a series of invertible transformations, $g = f_k \circ \cdots \circ f_1 \circ \mathrm{Pad}$, where $\mathrm{Pad}$ denotes padding a $n$-dimensional vector with $d - n$ zeros.

Just as for FOM and PIE, we model the base density $p_u(u)$ with an $n$-dimensional flow $h$, which maps $u$ to another latent variable $\tilde{u}$ with an associated tractable base density $p_{\tilde{u}}(\tilde{u})$. The induced probability density on the manifold is then given by

$$p_{\mathcal{M}}(x) = p_{\tilde{u}}(h^{-1}(g^{-1}(x))) \left| \det J_h(h^{-1}(g^{-1}(x))) \right|^{-1} \left| \det[J_g^T(g^{-1}(x)) J_g(g^{-1}(x))] \right|^{-\frac{1}{2}} . \quad (2)$$

This is the same as for FOM models, except with a learnable transformation $g$ rather than a prescribed, closed-form chart.

Sampling from an $\mathcal{M}$-flow is straightforward: one draws $\tilde{u} \sim p_{\tilde{u}}(\tilde{u})$ and pushes the latent variable forward to the data space as $u = h(\tilde{u})$ followed by $x = g(u) = f(u, 0)$, leading to data points on the manifold that consistently follow $x \sim p_{\mathcal{M}}(x)$.

As a final ingredient to the $\mathcal{M}$-flow approach, we add a prescription for evaluating arbitrary points $x \in X$, which may be off the manifold. $g$ maps from a low-dimensional latent space to the data space and is therefore a decoder. We define a matching encoder $g^{-1}$ as $f^{-1}$ followed by a projection to the $u$ component: $g^{-1} : X \mapsto U$, with $x \to g^{-1}(x) = \text{Proj}(f^{-1}(x))$ with $\text{Proj}(u, v) = u$. This extends the inverse of $g$ (which is so far only defined for $x \in \mathcal{M}$) to the whole data space $X$. Similar to an autoencoder,

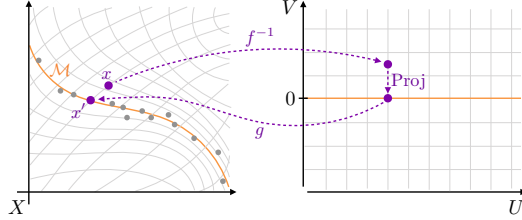

Figure 3: Sketch of how an $\mathcal{M}$-flow evaluates data on or off the learned manifold.

combining $g$ and $g^{-1}$ allows us to calculate a reconstruction error $\|x - x'\| = \|x - g(g^{-1}(x))\|$, which is zero if and only if $x \in \mathcal{M}$. Unlike for standard autoencoders, the encoder and decoder are exact inverses of each other as long as points on the manifold are studied.

For an arbitrary $x \in X$, an $\mathcal{M}$-flow thus lets us compute three quantities (see Fig. 3): the projection onto the manifold $x' = g(g^{-1}(x))$, which may be used as a denoised version of the input; the reconstruction error $\|x - x'\|$, which will be important to learn the manifold, but may also be useful for anomaly detection or out-of-distribution detection; and the likelihood on the manifold after the projection, $p_{\mathcal{M}}(x')$. In this way, $\mathcal{M}$-flows separate the distance from the data manifold and the density on the manifold—two concepts that easily get conflated in an ambient flow. $\mathcal{M}$-flows thus embrace ideas of energy-based models [9–11] for dealing with off-the-manifold issues, but still have a tractable, exact likelihood on the learned data manifold.

**Manifold-learning flows with separate encoder ($\mathcal{M}_e$-flow).** Finally, we introduce a variant of the $\mathcal{M}$-flow model where instead of using the inverse $f^{-1}$ followed by a projection as an encoder, we encode the data with a separate function $e : X \mapsto U$. This encoder is not restricted to be invertible or to have a tractable Jacobian, potentially increasing the expressiveness of the network. Just as in the $\mathcal{M}$-flow approach, for a given data point $x$ an $\mathcal{M}_e$-flow model returns a projected point onto the learned manifold, a reconstruction error, and the likelihood on the manifold evaluated after the projection. The added expressivity of this encoder comes at the price of potential inconsistencies between encoder and decoder, which the training procedure will have to try to penalize, exactly as for standard autoencoders.

**Probabilistic autoencoder (PAE).** How important is the invertibility of the transformation $f$ (and therefore $g$) in the $\mathcal{M}$-flow and $\mathcal{M}_e$-flow models? If we take the $\mathcal{M}_e$-flow model and replace $g$ with a decoder $g : U \mapsto X$ which is not required to be invertible, we arrive at an autoencoder model in which the latent space is modeled with a flow. This setup has recently been called probabilistic autoencoder (PAE) [16]. While relaxing the requirement of invertibility may make this generative model more expressive, it also loses the tractable density of the model.

**Manifolds with unknown dimensionality or nontrivial topology.** If the manifold dimension $n$ is not known a priori, it can be determined through cross-validation based on the reconstruction error or the performance on downstream tasks. Alternatively, for the PIE algorithm one could use trainable values of the base density variance $\varepsilon$ along each latent direction, allowing the model to learn the manifold dimensionality directly from the training data. If the manifold consists of multiple disjoint pieces, potentially with different dimensionality, a mixture model with multiple charts $g_i$ may be applicable [12].

# 3 Efficient training and evaluation

**Maximum likelihood is not enough.** Since the $\mathcal{M}$-flow density is tractable, maximum likelihood is an obvious candidate for a training objective. However, the situation is more subtle as the $\mathcal{M}$-flow model describes the density *after projecting onto the learned manifold*. The definition of the data variable in the likelihood hence depends on the weights $\phi_f$ of the manifold-defining transformation $f$, and a comparison of naive likelihood values between different configurations of $\phi_f$ is meaningless. Instead of thinking of a likelihood function $p(x|\phi_f, \phi_h)$, where $\phi_h$ are the weights defining $h$, it is instructive to think of a family of likelihood functions $p_{\phi_f}(x|\phi_h)$ parameterized by the different $\phi_f$.

Training $\mathcal{M}$-flows by simply maximizing the naive likelihood $p(x|\phi_f, \phi_h)$ therefore does not incentivize the network to learn the correct manifold. As an extreme example, consider a model manifold that is perpendicular to the true data manifold. Since this configuration allows the $\mathcal{M}$-flow to project all points to a small region of high density on the model manifold, this pathological configuration may lead to a high naive likelihood value. We demonstrate this issue with a concrete example in Sec. 3.A of the supplementary material.

A second challenge is the computational efficiency of evaluating the $\mathcal{M}$-flow density in Eq. (2). While this quantity is in principle tractable, it cannot be computed as efficiently as the likelihood of an ambient flow. The underlying reason is that since the Jacobian $J_g$ is not square, it is not obvious how the determinant $\det J_g^T J_g$ can be decomposed further, at least when we compose an $\mathcal{M}$-flow out of the typical elements of ambient flows like coupling layers. Evaluating the $\mathcal{M}$-flow density then requires the computation of all entries of the Jacobians of the individual transformation. While this cost can be reasonable for the evaluation of a limited number of test samples, it can be prohibitively expensive during training. Since the computational cost grows with increasing data dimensionality $d$, training by maximizing $\log p_{\mathcal{M}}$ does not scale to high-dimensional problems.

**Separate manifold and density training (M/D).** We can solve both problems at once by separating the training into two phases. In the *manifold phase*, we update only the parameters of $f$, which through a level set also define the manifold $\mathcal{M}$ and the chart $g$. Similarly to autoencoders, we minimize the reconstruction error from the projection onto the manifold, $\|x - g(g^{-1}(x))\|$. For the $\mathcal{M}_e$-flow model, the parameters of the encoder $e$ are also updated during this phase. In the *density phase*, we update only the parameters of $h$ by maximum likelihood. $h$ only affects the density $p_u$, we are thus keeping the manifold fixed during this phase.

Such a training procedure is not prone to the gradient flow aligning the manifold with pathological configurations, as we demonstrate in Sec. 3.A of the supplementary material. Moreover, training only $h$ by maximum likelihood does not require computing the expensive terms in the model likelihood. The loss in the density phase is given by

$$L[h] = -\frac{1}{n} \sum_x \left( \log p_{\tilde{u}}(h^{-1}(u)) - \log \det J_h(h^{-1}(u)) - \frac{1}{2} \log \det[J_g^T(u)J_g(u)] \right) \qquad (3)$$

with $u = g^{-1}(x)$. Only the last term is expensive to evaluate, but it does not depend on the parameters of $h$ and does not contribute to the gradient updates in this phase! We can therefore train the parameters of $h$ by minimizing only the first two terms, which can be evaluated efficiently. The two phases can be scheduled sequentially or in an alternating schedule, see Sec. 3.B of the supplementary material.

**Likelihood evaluation.** For high-dimensional data, evaluating the likelihood in Eq. (2) can become so expensive that even the likelihood evaluations at test time must be limited. In Sec. 3.C of the supplementary material we discuss several approximate inference techniques that may reduce this computational cost. We also argue that this issue can be solved efficiently and exactly in the common case where the density (but not the manifold) is conditional on some model parameters $\theta$ and the downstream goal is inferring these model parameters $\theta$. In this case, the $\mathcal{M}$-flow setup enables the fast and exact computation of likelihood *ratios* and MCMC acceptance probabilities.

# 4 Experiments

A common metric for flow-based models is the likelihood evaluated on a test set, but such a comparison is not meaningful in our context. Since the $\mathcal{M}$-flow variants evaluate the likelihood after

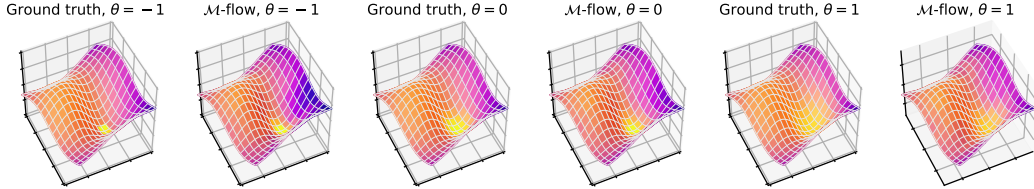

Figure 4: Ground truth and $\mathcal{M}$-flow model for the mixture model on a polynomial surface.

projecting to a learned manifold, the data variable in the likelihood is different for every model and the likelihoods of different models may not even have the same units. Instead, we analyze the performance through the generative mode, the quality of the manifold, or the performance on downstream inference tasks, depending on the experiment. In Sec. 4 of the supplementary material we describe our setup in detail and provide additional experiments, more results, and exhaustive discussions.

**Architectures.** We compare $\mathcal{M}$-flow and $\mathcal{M}_e$-flow models to AF, PIE, and PAE baselines. All models are based on rational-quadratic neural spline flows [17]. For tabular datasets, we construct transformations $f$ and $h$ by alternating coupling layers with either random permutations or invertible linear transformations, using between 20 and 35 coupling layers depending on the dataset. For image data, $f$ is based on a multi-scale architecture [5] with between 20 and 28 coupling layers across four levels interspersed with actnorm layers and $1 \times 1$ convolutions, closely following Refs. [17, 18]. In the $\mathcal{M}$-flow, $\mathcal{M}_e$-flow, and PIE models we apply two additional invertible linear layers to a subset of the channels before projecting to the manifold coordinates $u$; this gives the models the freedom to align the learned manifold with features across different scales. For the transformation $h$ we use the same setup as for tabular data. While the inherently different architectures between the manifold-aware models and the manifold-agnostic AF make it hard to match the configurations exactly, we try to ensure fairness by using the same overall number of coupling layers and matching as many of the other hyperparameters as possible.

**Mixture model on a polynomial surface.** As a first synthetic example we consider a two-dimensional manifold embedded in $\mathbb{R}^3$ defined by $x = R \left( z_0, z_1, f(z) \right)^T$. Here $z = (z_0, z_1)^T$ is a vector of two latent variables that parameterize the manifold, $f(z) = \exp(-0.1\|z\|) \sum_{i,j} a_{ij} z_0^i z_1^j$, and the rotation matrix $R$ as well as the coefficients $a_{ij}$ are given in the supplementary material. We generate training and test data on this manifold from a mixture model of two Gaussians in the latent space, which is conditional on a model parameter $\theta \in [-1, 1]$.

We train models on $10^5$ training samples and evaluate them on four metric in Tbl. 2. Samples generated from the flows are judged based on the mean distance to the true data manifold. The quality of the learned manifold in the PIE, $\mathcal{M}$-flow, and $\mathcal{M}_e$-flow models is estimated with the reconstruction error when projecting test samples to the manifold. As an inference task we compute the posterior over $\theta$ given some observed data, using MCMC samplers based on the different model likelihoods; we report the maximum mean discrepancies (MMD) between the true posterior and the approximate posteriors [19]. Finally, we evaluate out-of-distribution (OOD) detection by comparing the distribution of log likelihood (AF, PIE) or reconstruction error ($\mathcal{M}$-flow, $\mathcal{M}_e$-flow, PAE) between a test sample and an OOD sample.

In all metrics except OOD detection, manifold-learning flows provide the best results. $\mathcal{M}$-flow and

| Model | Manifold distance | Reconstruction error | Posterior MMD | OOD AUC |
|---|---|---|---|---|
| AF | 0.005 | – | 0.071 | **0.990** |
| PIE (original) | 0.035 | 1.278 | 0.131 | 0.933 |
| PIE (uncond. manifold) | 0.006 | 1.253 | 0.075 | 0.972 |
| PAE | **0.002** | **0.002** | – | **0.990** |
| $\mathcal{M}$-flow (alternating M/D) | **0.002** | **0.003** | 0.020 | 0.986 |
| $\mathcal{M}$-flow (sequential M/D) | 0.009 | 0.013 | **0.017** | 0.961 |
| $\mathcal{M}_e$-flow (alternating M/D) | **0.003** | **0.003** | 0.030 | 0.985 |
| $\mathcal{M}_e$-flow (sequential M/D) | **0.002** | **0.002** | **0.007** | **0.987** |

Table 2: Results for the mixture model on a polynomial surface, showing the median out of 5 runs. The best results, generally consistent with each other within the variance observed in the five runs, are shown in bold.

| Model (algorithm) | Closure | Reco. error | Log posterior |
|---|---|---|---|
| AF | **0.0019** ± 0.0001 | – | −3.94 ± 0.87 |
| PIE (original) | 0.0023 ± 0.0001 | 2.054 ± 0.076 | −4.68 ± 1.56 |
| PIE (unconditional manifold) | 0.0022 ± 0.0001 | 1.681 ± 0.136 | −1.82 ± 0.18 |
| PAE | 0.0073 ± 0.0001 | 0.052 ± 0.001 | – |
| $\mathcal{M}$-flow | 0.0045 ± 0.0004 | **0.012** ± 0.001 | −1.71 ± 0.30 |
| $\mathcal{M}_e$-flow | 0.0046 ± 0.0002 | 0.029 ± 0.001 | −1.44 ± 0.34 |
| AF (SCANDAL) | 0.0565 ± 0.0059 | – | −0.40 ± 0.09 |
| PIE (original, SCANDAL) | 0.1293 ± 0.0218 | 3.090 ± 0.052 | 0.03 ± 0.17 |
| PIE (uncond. manifold, SCANDAL) | 0.1019 ± 0.0104 | 1.751 ± 0.064 | **0.23** ± 0.05 |
| PAE (SCANDAL) | 0.0323 ± 0.0010 | 0.053 ± 0.001 | – |
| $\mathcal{M}$-flow (SCANDAL) | 0.0371 ± 0.0030 | **0.011** ± 0.001 | 0.11 ± 0.04 |
| $\mathcal{M}_e$-flow (SCANDAL) | 0.0291 ± 0.0010 | 0.030 ± 0.002 | **0.14** ± 0.09 |
| Likelihood ratio estimator (ALICES) | – | – | 0.05 ± 0.05 |

Table 3: Results for the particle physics dataset, showing the mean between at least 5 runs after removing the best and worst run.

$\mathcal{M}_e$-flow samples are closest to the true data manifold, most faithfully reconstruct test samples after projecting them to the learned manifold, and clearly outperform the AF and PIE baselines when it comes to inference on $\theta$. They are on par with PAE, which does not have a tractable density, showing that the restriction to an invertible decoder does not pose a significant restriction to the expressiveness of the model. We show the ground truth data manifold and one $\mathcal{M}$-flow model in Fig. 4.

**Lorenz attractor.** The Lorenz system is a three-dimensional, non-linear, deterministic system in which $x \in \mathbb{R}^3$ evolves with time under the equations $\frac{dx_0}{dt} = \sigma(x_1 - x_0)$, $\frac{dx_1}{dt} = x_0(\rho - x_2) - x_1$, and $\frac{dx_2}{dt} = x_0 x_1 - \beta x_2$. For certain parameter choices and initial conditions it has chaotic solutions that tend to the *Lorenz attractor*, a strange attractor with Hausdorff dimension of approximately 2.06 [20]. We train an $\mathcal{M}$-flow model to learn the invariant probability density of the Lorenz at-

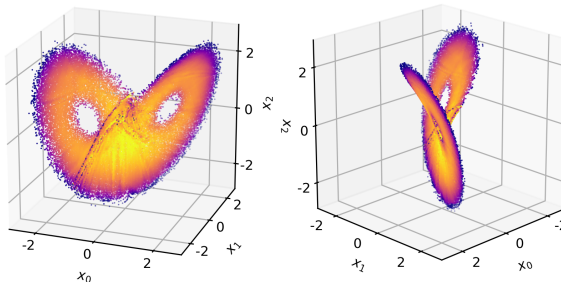

Figure 5: Manifold and invariant density of the Lorenz attractor learned by an $\mathcal{M}$-flow model from two different perspectives.

tractor on a two-dimensional manifold. The learned manifold and density, shown in Fig. 5, are plausible; the $\mathcal{M}$-flow is even able to describe the nontrivial disconnected branches of the attractor.

**Particle physics.** Next, we consider a real-world problem from particle physics. Proton-proton collisions at the Large Hadron Collider experiments lead to observations $x \in \mathbb{R}^{40}$. The likelihood $p^*(x|\theta)$ is conditional on 3 constants of nature $\theta$ and is only known implicitly through a simulator, though from the laws of particle physics we know that the data must be restricted to some 14-dimensional manifold embedded in the 40-dimensional data space. Given an observation $\{x_{\text{obs}}\}$, the goal is inference on $\theta$, a setting known as simulation-based (or likelihood-free) inference [21].

We train flow models on $10^6$ training samples generated from the simulator to learn the likelihood function. In addition to the usual training schemes we consider models trained with the SCANDAL loss [22], in which additional information from the simulator is leveraged to make the training more sample efficient. As an additional baseline we consider an estimator of the likelihood *ratio* function trained with the ALICES method [23]; while likelihood ratio estimators are known to provide a strong baseline for inference, they lose the ability to generate data [21, 24].

Samples generated from the flows are first evaluated on a domain-specific closure test, with a closure of 0 being best and a closure of 1 corresponding to the product of the marginal densities (or a random reshuffling of all features across samples). The learned manifolds are again evaluated on the reconstruction error when projecting test samples. Finally, we use an MCMC sampler based on the different model likelihoods and kernel density estimation to compute the posterior for different synthetic observations. Since the true posterior is intractable, we assess the inference quality through the log posterior evaluated at the true parameter point used to generate the observed samples.

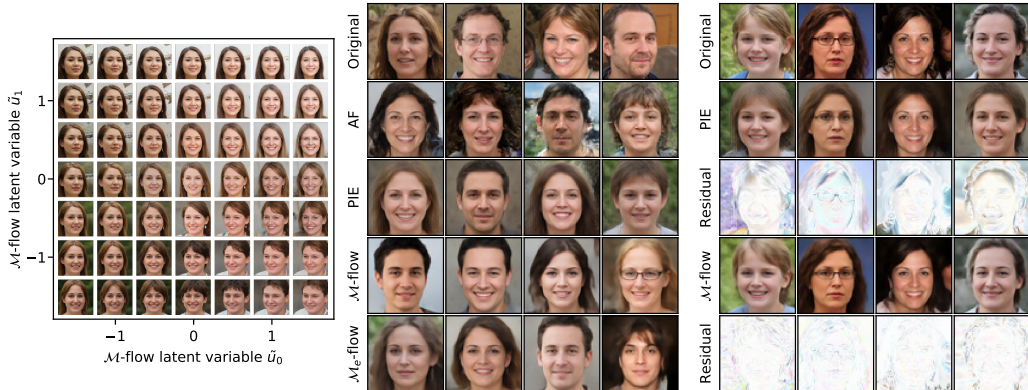

Figure 6: Left: 2-D manifold learned by an $\mathcal{M}$-flow. Middle: uncurated flow samples for the 64-D image manifold dataset. Right: 64-D manifold test samples, their projection to the learned manifold, and residuals.

The results in Tbl. 3 show that the AF models produce the most realistic samples according to the closure test, but do not result in reliable inference results: pure likelihood-based training seems to incentivize learning the overall data distribution more than figuring out the subtle effects of the parameters of interest. Our improved PIE version with an unconditional manifold and the $\mathcal{M}$-flow/$\mathcal{M}_e$-flow models perform better on the inference task. $\mathcal{M}$-flow/$\mathcal{M}_e$-flow models define higher-quality manifolds with lower projection errors than PIE and PAE. SCANDAL training reduces the sample quality as measured by the closure test, but substantially improves the quality of inference, in some cases even outperforming the baseline likelihood ratio estimator.

**Images.** Finally, we study three image datasets. The first two consist of synthetic images that populate an $n$-dimensional manifold. We generate these with a Style-GAN2 [25] model trained on the FFHQ dataset [26], sampling $n$ of the GAN latent variables while keeping all

| Model | FID scores | | | Log posterior |
|---|---|---|---|---|
| | $n = 2$ | $n = 64$ | CelebA | $n = 64$ |
| AF | $58.3 \pm 1.5$ | $24.0 \pm 0.0$ | $\mathbf{33.6} \pm 0.2$ | $0.17 \pm 1.18$ |
| PIE | $139.5 \pm 5.0$ | $32.2 \pm 0.8$ | $75.7 \pm 5.1$ | $-6.40 \pm 1.54$ |
| $\mathcal{M}$-flow | $43.9 \pm 0.2$ | $\mathbf{20.8} \pm 0.5$ | $37.4 \pm 0.2$ | $\mathbf{2.67} \pm 0.27$ |
| $\mathcal{M}_e$-flow | $\mathbf{43.5} \pm 0.2$ | $23.7 \pm 0.2$ | $35.8 \pm 0.4$ | $1.81 \pm 0.70$ |

Table 4: FID scores and inference metrics for the $n$-dimensional StyleGAN image manifolds and CelebA. We show the mean out of at least 3 runs.

others fixed. We consider a dataset with $n = 2$ and $10^4$ training samples as well as one with $n = 64$ and $2 \cdot 10^4$ training samples; for the latter the distribution of images depends on a model parameter $\theta$. In addition, we use the real-world CelebA-HQ dataset [26]. Here the existence of a data manifold and its dimension are not known, for the $\mathcal{M}$-flow and $\mathcal{M}_e$-flow models we use $n = 512$. All images are downsampled to a resolution of $64 \times 64$.

In Tbl. 4 we evaluate the flow models based on the Fréchet Inception Distance (FID score) [27, 28]. For $n = 64$, we also evaluate the quality of inference on $\theta$ given an observed dataset, reporting the log posterior at the true parameter point $\theta^*$. Both $\mathcal{M}$-flow and $\mathcal{M}_e$-flow models perform better than the AF and PIE baselines on the StyleGAN image manifold datasets, though slightly worse than the AF models on CelebA. This may point to a suboptimal choice of manifold dimension $n$; in Sec. 4.F of the supplementary material we study the dependence of the $\mathcal{M}$-flow performance on $n$. In all experiments, $\mathcal{M}$-flows and $\mathcal{M}_e$-flows find higher-quality manifolds than PIE models. In Fig. 6 we show an image manifold learned by an $\mathcal{M}$-flow model, samples generated from different flows, and projections of test samples onto learned image manifolds.

## 5 Related work

Our work is closely related to a number of different probabilistic and generative models, many of which were discussed in Sec. 2. In addition, manifold learning is its own research field with a rich set of methods [29], though these typically do not model the data density on the manifold and thus do not serve quite the same purpose as the models discussed in this paper. Here we want to draw attention to two particularly closely related works and describe how our approach differs from them.

**Injective flows.**   Relaxed injective probability flows [30], which appeared while this paper was in its final stages of preparation, are similar to our $\mathcal{M}_e$-flows. Instead of using invertible flow transformations, they enforce the invertibility of the decoder $g$ by bounding the norm of the Jacobian of an otherwise unrestricted transformation. While this makes the transformation locally invertible, it does not eliminate the possibility of multiple points in latent space pointing to the same point in data space. Furthermore, the inverse of $g$ and the likelihood are not tractable for unseen data points, limiting the usefulness for inference tasks. The approaches also differ in the treatment of points off the learned manifold and the training; we discuss these aspects in Sec. 5 of the supplementary material.

**Pseudo-invertible encoder.**   We define and discuss pseudo-invertible encoders (PIE) [15] in Sec. 2 and use them as a baseline in our experiments. The key difference to our $\mathcal{M}$-flow setup is that the PIE density is not restricted to the manifold and, similar to an ambient flow, permeates the ambient data space. The proposed generative mode is restricted to the manifold, but inconsistent with the density, as we argue in detail in Sec. 2.C of the supplementary material.

# 6   Conclusions

In this work we introduced manifold-learning flows ($\mathcal{M}$-flows), a new type of generative model that combines aspects of normalizing flows, autoencoders, and GANs. $\mathcal{M}$-flows describe data with a tractable probability density over a lower-dimensional manifold embedded in data space. Both the manifold and the density are learned from data. We identified a subtlety in the naive interpretation of the density of such models, argued that they should not be trained by naive maximum likelihood alone, and addressed this issue with the stable and efficient M/D training strategy.

In a first suite of experiments ranging from simple pedagogical examples to a real-world physics example to image datasets, we demonstrated how this approach lets us learn the data manifold and a probability density on it. $\mathcal{M}$-flows learned manifolds of a higher quality than PIE baselines, and performed better than ambient flow and PIE models on most downstream inference tasks and many of the generative metrics we considered.

Problems in which data populate a lower-dimensional manifold embedded in a high-dimensional data space are almost everywhere. The manifold structure may be particularly explicit in some scientific problems, while the success of GANs on numerous datasets is testament to the presence of low-dimensional data manifolds in other domains. Manifold-learning flows may help us unify generative and inference tasks in a way that is tailored to the structure of the data.

## Acknowledgements

We would like to thank Jens Behrmann, Kyunghyun Cho, Jack Collins, Jean Feydy, Siavash Golkar, Michael Kagan, Dimitris Kalatzis, Gilles Louppe, George Papamakarios, Merle Reinhart, Frank Rösler, John Tamanas, Antoine Wehenkel, and Andrew Wilson for useful discussions. We are grateful to Conor Durkan, Artur Bekasov, Iain Murray, and George Papamakarios for publishing their excellent neural spline flow codebase [17], which we used extensively in our analysis. Similarly, we want to thank George Papamakarios, David Sterratt, and Iain Murray for publishing their Sequential Neural Likelihood code [31], parts of which were used in the evaluation steps in our experiments. We are grateful to the authors and maintainers of DELPHES 3 [32], GEOMLOSS [33], JUPYTER [34], MADGRAPH5_AMC [35], MADMINER [36], MATPLOTLIB [37], NUMPY [38], PYTHIA8 [39], PYTORCH [40], PYTORCH-FID [41], SCIKIT-LEARN [42], and SCIPY [43].

## Funding disclosure

We are grateful for the support of the National Science Foundation under the awards ACI-1450310, OAC-1836650, and OAC-1841471, as well as by the Moore-Sloan data science environment at NYU. This work was supported in part through the NYU IT High Performance Computing resources, services, and staff expertise; through the NYU Courant Institute of Mathematical Sciences; and by the Scientific Data and Computing Center at Brookhaven National Laboratory.

## Broader impact

Manifold-learning flows have the potential to improve the efficiency with which scientists extract knowledge from large-scale experiments. Many phenomena have their most accurate description in terms of complex computer simulations which do not admit a tractable likelihood. In this common case, normalizing flows can be trained on synthetic data and used as a surrogate for the likelihood function, enabling high-quality inference on model parameters [21]. When the data have a manifold structure, manifold-learning flows may improve the quality and efficiency of this process further and ultimately contribute to scientific progress. We have demonstrated this with a real-world particle physics dataset, though the same technique is applicable to fields as diverse as neuroscience, systems biology, and epidemiology.

All generative models carry a risk of being abused for the generation of fake data that are then masqueraded as real documents. This danger also applies to manifold-learning flows. While manifold-learning flows are currently far away from being able to generate realistic high-resolution images, videos, or audio, this concern should be kept in mind in the long term.

Finally, the models we trained on image datasets of human faces clearly lack diversity. They reproduce and reinforce the biases inherent in the training data. Before using such (or other) models in any real-life application, it is crucial to understand, measure, and mitigate such biases.

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
