[Supplementary Material]

# Flows for simultaneous manifold learning and density estimation: Supplementary material

**Johann Brehmer**[a,b,1] **and Kyle Cranmer**[a,b]

[a]Center for Data Science, New York University, USA; [b]Center for Cosmology and Particle Physics, New York University, USA

October 19, 2020

We introduce manifold-learning flows ($\mathcal{M}$-flows), a new class of generative models that simultaneously learn the data manifold as well as a tractable probability density on that manifold. Combining aspects of normalizing flows, GANs, autoencoders, and energy-based models, they have the potential to represent datasets with a manifold structure more faithfully and provide handles on dimensionality reduction, denoising, and out-of-distribution detection. We argue why such models should not be trained by maximum likelihood alone and present a new training algorithm that separates manifold and density updates. In a range of experiments we demonstrate how $\mathcal{M}$-flows learn the data manifold and allow for better inference than standard flows in the ambient data space.

**Fig. 1.** Sketch of how a standard normalizing flow in the ambient data space (left, orange surface) and an $\mathcal{M}$-flow (right, purple) model data (black dots).

## Contents

## Preface to the supplementary material

This document contains additional details, in-depth discussions, and more results that did not fit into the main paper. Rather than providing a series of appendices, we tried to turn it into a substantially extended version of the paper that can be read on its own, without requiring jumping back and forth between two documents.

## 1. Introduction

Inferring a probability distribution from example data is a common problem that is increasingly tackled with deep generative models. Generative adversarial networks (GANs) (1) and variational autoencoders (VAEs) (2) are both based on a lower-dimensional latent space and a learnable mapping from that to the data space. In essence, these models describe a lower-dimensional data manifold embedded in the data space. While they allow for efficient sampling, their probability density (or likelihood) is intractable, leading to a challenge for training and limiting their usefulness for inference tasks. On the other hand, normalizing flows (3–6) are based on a latent space with the same dimensionality as the data space and a diffeomorphism; their tractable density permeates the full data space and is not restricted to a lower-dimensional surface.

The flow approach may be unsuited to data that do not populate the full ambient data space they natively reside in, but are restricted to a lower-dimensional manifold (7). Normalizing flows are by construction not able to represent such a structure exactly, instead they learn a smeared-out version with support off the manifold. We illustrate this in the left panel of Figure 1, where the black dots represent 2D data populating a 1D manifold and the orange surface sketches the density learned by a normalizing flow. In addition, the requirement of latent spaces with the same dimension as the data space increases the memory footprint and computational

cost of the model. While flows have been generalized from Euclidean spaces to Riemannian manifolds (8), this approach has so far been limited to the case where the chart for the manifold is prescribed.

We introduce *manifold-learning flows* ($\mathcal{M}$-flows): normalizing flows based on an injective, invertible map from a lower-dimensional latent space to the data space. $\mathcal{M}$-flows simultaneously learn the shape of the data manifold, provide a tractable bijective chart, and learn a probability density over the manifold, as sketched in the right panel of Figure 1. When evaluating the model, the input (which may be off the manifold) is first projected onto the manifold and the model returns both the distance from the manifold as well as the density on the manifold after the projection.

The $\mathcal{M}$-flow approach marries aspects of normalizing flows, GANs, and autoencoders. Compared to flows on prescribed manifolds, this approach relaxes the requirement of knowing a closed-form expression for the chart from latent variables to the data manifold and instead learns the manifold from data. In contrast to GANs and VAEs, it not only provides an exact tractable likelihood over the data manifold, but also a prescription for how to treat points off the manifold. In contrast to standard autoencoders, it is a probabilistic model with a generative mode and tractable density. Similar to invertible autoencoders (9), $\mathcal{M}$-flows ensure that for data points on the manifold the encoder and decoder are the inverse of each other. They can also be seen as regularized autoencoders (10, 11). Compared to standard flow-based generative models, $\mathcal{M}$-flows offer four advantages:

- $\mathcal{M}$-flows may more accurately approximate the true data distribution, avoiding probability mass off the data manifold. This in turn could lead to performance gains in inference and generative tasks.

- The model architecture naturally allows one to model a conditional density that lives on a fixed manifold. This should improve data efficiency in such situations as it is ingrained in the architecture and does not need to be learned.

- The lower-dimensional latent space reduces the complexity of the model, allowing us to use more expressive transformations or scale to higher-dimensional data spaces within a given computational budget.

- The projection onto the data manifold provides dimensionality reduction and denoising capabilities. The distance to the manifold may also be useful to detect out-of-distribution samples.

The $\mathcal{M}$-flow model embraces the idea of energy-based models (12–14) for dealing with off-the-manifold issues through a non-probabilistic distance measure, while retaining a tractable density on the data manifold. Similarly, we can link it to the development of adversarial objectives for GANs: the original GAN setup (1), in which a generator is pitted against a discriminator, corresponds to training based on a proxy for the likelihood ratio. Off the data manifold this density ratio is not well-defined, which makes the training challenging. Wasserstein GANs (15) address this issue by measuring distances between two data manifolds in feature space. Similarly, the likelihood of normalizing flows is not appropriate when data populates a lower-dimensional manifold; $\mathcal{M}$-flows augment flows with a distance measure in feature space to measure closeness to the data manifold.

Training an $\mathcal{M}$-flow model faces two challenges. First, maximum likelihood is not enough: we will demonstrate that the training dynamics for naive likelihood-based training may not lead to a good estimate of the manifold and the density on it. Second, evaluating the $\mathcal{M}$-flow density can be computationally expensive. We will discuss several new training strategies that solve these challenges. In particular, we introduce a new training scheme with separate manifold and density updates, which allows for a computationally efficient training of $\mathcal{M}$-flows and incentivizes both good manifold quality and good density estimation on the manifold.

We begin with a broad discussion of the notion of data manifolds in different generative models and introduce manifold-learning flows in Section 2. In Section 3 we discuss pitfalls when training $\mathcal{M}$-flows and introduce training strategies that can overcome these challenges. In Section 4 we demonstrate $\mathcal{M}$-flow in experiments. We comment on related work in Section 5 before summarizing the results in Section 6. The code used in our study is available at http://github.com/johannbrehmer/manifold-flow.

## 2. Generative models and the data manifold

Consider a data-generating process that draws samples $x \in \mathcal{M}^* \subset X = \mathbb{R}^d$ according to $x \sim p^*(x)$, where $\mathcal{M}^*$ is a $n$-dimensional Riemannian manifold embedded in the $d$-dimensional data space $X$ and $n < d$. We consider the two problems of estimating the density $p^*(x)$ as well as the manifold $\mathcal{M}^*$ given some training samples $\{x_i\} \sim p^*(x)$. We will later extend our models with a projection to the manifold so that they can also handle problems in which the data is only approximately restricted to a manifold.

In the following we will discuss which types of generative models address which parts of this problem and generally discuss the relation between the data manifold and various classes of generative models. In the process we will also introduce the new manifold-learning flows ($\mathcal{M}$-flows). We distinguish between three different groups of models:

1. manifold-free models defined in the ambient space $X$,

2. models for an explicitly prescribed manifold, and

3. models that learn an unknown manifold.

In this discussion we rely on a few simplifying assumptions. We treat the manifold as topologically equivalent to $\mathbb{R}^n$; in particular, we assume that it is connected and can be described by a single chart. We also assume that the dimensionality $n$ of the manifold is known. In Section 2.D we will discuss how these requirements can be lifted.

To facilitate a straightforward comparison, we will describe all generative models in terms of two vectors of latent variables $u \in U$ and $v \in V$, where $U = \mathbb{R}^n$ is the latent space that maps to the learned manifold $\mathcal{M}$, i.e. the coordinates of the manifold. $V = \mathbb{R}^{d-n}$ parameterizes any remaining latent variables, representing the directions "off the manifold".

In Figure 2 we sketch the setup of the different models. In Table 1 we summarize some of their properties.

## A. Manifold-free models: Ambient flows.

**Ambient flow (AF).** In these conventions, a standard Euclidean normalizing flow (6) in the ambient data space is a diffeomorphism

$$f : U \times V \mapsto X$$
$$u, v \to f(u, v) \tag{1}$$

together with a tractable base density (such as a multivariate unit Gaussian) $p_{uv}(u, v)$. According to the change-of-variable formula, the density in $X$ is then given by

$$p_x(x) = p_{uv}(f^{-1}(x)) \left| \det J_f(f^{-1}(x)) \right|^{-1} , \tag{2}$$

where $J_f$ is the Jacobian of $f$, a $d \times d$ matrix. $f$ is usually implemented as a neural network with certain constraints that make $\det J_f$ efficient to compute. In the generative mode, flows sample $u$ and $v$ from their base densities and apply the transformation $x = f(u, v)$, leading to samples $x \sim p_x(x)$.

There is typically no difference between $u$ and $v$. While some models employ multi-scale architectures where some latent variables have more transformations applied to them than others (5), there is no explicit incentive for the network to align these directions in the latent space with coordinates on the data manifold and off-the-manifold directions, respectively. This model therefore has no notion of a data manifold, they only describe regions of varying probability density in the overall ambient data space. We will therefore refer to it as *ambient flow* (AF).

## B. Flows on a prescribed manifold.

**Flow on a manifold (FOM).** When a chart (or an atlas of multiple charts) for the manifold is known a priori, one can construct a flow on this manifold (8). If a diffeomorphism

$$g^* : U \mapsto \mathcal{M}^* \subset X$$
$$u \to g^*(u) \tag{3}$$

is the sole chart for the manifold, the density on the manifold is given by

$$p_{\mathcal{M}^*}(x) = p_u(g^{*-1}(x)) \left| \det[J_g^T(g^{*-1}(x)) J_g(g^{*-1}(x))] \right|^{-\frac{1}{2}} , \tag{4}$$

where $J_g$ is the Jacobian of $g^*$, an $n \times d$ matrix. The latent variables $u$ are the coordinates of the manifold. The density $p_u(u)$ in this coordinate space can then be modeled with a regular normalizing flow in $n$ dimensions with a learnable diffeomorphic transformation

$$h : \tilde{U} \mapsto U$$
$$\tilde{u} \to h(\tilde{u}) \tag{5}$$

and a base density $p_{\tilde{u}}(\tilde{u})$. Then

$$p_u(u) = p_{\tilde{u}}(h^{-1}(u)) \left| \det J_h(h^{-1}(u)) \right|^{-1} , \tag{6}$$

where $J_h$ is the Jacobian of $h$.

Sampling from such a flow is straightforward and consists of drawing from the base density and transforming the variable with $h$ and then $g^*$. Depending on the choice of chart, the model likelihood in Eq. (4) can be evaluated efficiently, and the model is by construction limited to the true manifold. This approach has been worked out for spheres and tori of arbitrary dimension (16), for hyperbolic manifolds (17), as well as for a problem in theoretical physics where the manifold consists of a particular product of $U(1)$ groups (18).

**Fig. 2.** Schematic relation between data $x$ and various latent variables $u, v, \tilde{u}$ in the different generative models discussed in Section 2. Red arrows represent learnable transformations, while black arrows stand for fixed transformations. Solid lines show invertible bijections, dashed lines denote injections that are invertible within their image, and dotted lines show unrestricted transformations that may be neither injective nor invertible.

## C. Learning the manifold: From GANs to $\mathcal{M}$-flows.

**Generative adversarial network (GAN).** GANs map an $n$-dimensional latent space to the data space,

$$g : U \mapsto \mathcal{M} \subset X$$
$$u \to g(u) . \tag{7}$$

Here $g$ is a learnable map like a deep neural network rather than a prescribed closed-form chart. This map is neither restricted to be invertible nor injective: there can be multiple $u$ that correspond to the same data point $x$. Therefore $g$ is not a chart and the image of this transformation not necessarily a Riemannian manifold, though this distinction is not our focus and we will simply call this subset a manifold.

While the lack of restrictions on $g$ increases the expressivity of the neural network, it also makes the model density intractable. This drawback has two immediate consequences. First, GANs have to be trained adversarially as opposed to by maximum likelihood. Second, despite their built-in manifold-like structure GANs are neither well-suited for inference tasks that require to evaluate the model density nor for manifold learning tasks.[*] Finally, note that in conditional GANs both

---

[*] Reference (19) introduces a method that allows to calculate the GAN density at least approximately, though this approach neglects the possibility of multiple $u$ pointing to the same $x$. PresGANs (20) add a noise term to the generative procedure, similar to a VAE, as well as a numerical method to evaluate the model density approximately using importance sampling.

| Model | Manifold | Chart | Generative mode | Tractable density | Restricted to manifold |
|---|---|---|---|---|---|
| Ambient flow (AF) | no manifold | ✗ | ✓ | ✓ | ✗ |
| Flow on manifold (FOM) | prescribed | ✓ | ✓ | ✓ | ✓ |
| Generative adversarial network (GAN) | learned | ✗ | ✓ | ✗ | ✓ |
| Variational autoencoder (VAE) | learned | ✗ | ✓ | only ELBO | (✗) |
| Pseudo-invertible encoder (PIE) | learned | ✓ | ✓ | ✓ | (✗) |
| Slice of PIE | learned | ✓ | ✗ | up to normalization | ✓ |
| Probabilistic autoencoder (PAE) | learned | ✗ | ✓ | ✗ | ✓ |
| Manifold-learning flow ($\mathcal{M}$-flow) | learned | ✓ | ✓ | ✓ (may be slow) | ✓ |
| Manifold-learning flow with sep. encoder ($\mathcal{M}_e$-flow) | learned | ✓ | ✓ | ✓ (may be slow) | ✓ |

**Table 1.** Generative models for data that populate a lower-dimensional manifold. We differentiate the models by whether they have a prescribed or learned internal notion of the data manifold, whether they provide access to a diffeomorphic chart of that manifold, if they allow us to generate samples, whether they have a tractable density, and whether the model density is actually restricted to the manifold (as opposed to the full ambient data space). In the last column, parantheses (✗) mean that an alternative sampling procedure can generate data just from the manifold, but that this sampling process does not follow the model density.

the shape of the manifold as well as the implicit density on it generally depend on the variables $\theta$ being conditioned on.[†]

**Variational autoencoder (VAE).** Variational autoencoders also map a lower-dimensional latent space $U$ to the data space, but instead of a deterministic function $x = g(u)$ they use a stochastic decoder $p(x|u)$. The marginal density

$$p(x) = \int \mathrm{d}u\, p(x|u)p(u) \qquad [8]$$

of the model therefore extends off the manifold into the whole space $X$. This marginal density itself is intractable, though there is a variational lower bound (the ELBO) for it that is commonly used as a training objective.

Nevertheless, the lower-dimensional latent space of a VAE is often associated with a learned data manifold. Often only the final step in the decoder is stochastic, for instance as a Gaussian density in data space where the mean is a learned function of the latent variables. Then one can define an alternative generative mode by using this mean instead of sampling from the Gaussian, replacing the stochastic decoder with a deterministic one. In this way the generated samples are restricted to a lower-dimensional subset $\mathcal{M} \in X$. While not strictly a manifold, for all practical purposes it is equivalent to one. However, generating in this mode does not correspond to sampling from $p(x)$, which was used to train the model.

**Pseudo-invertible encoder (PIE).** One way to give ambient flows a notion of a (learnable) manifold is to treat some of the latent variables differently from others and rely on the training to align one class of latent variables with the manifold coordinates and the other class of latent variables with the off-the-manifold directions. This is the essential idea behind the pseudo-invertible encoder (PIE) architecture (21). Its basic setup is given by the flow transformation of Eq. (1) and the flow density in Eq. (2). The key difference is that in PIE one chooses different base densities for the latent variables $u$, which are designated to represent the coordinates on the manifold, and $v$, which should learn the off-the-manifold directions in latent space: the base density $p_u(u)$ is modeled with an $n$-dimensional Euclidean flow, i.e. a transformation $h$ that maps it to another latent variable $\tilde{u}$ associated with a standard base density such as a unit Gaussian. The off-the-manifold

base density $p_v(v)$ is chosen such that it sharply peaks around 0, for instance as a Gaussian with a small variance $\varepsilon^2 \ll 1$ in each direction.

For sufficiently flexible transformations, this architecture has the same expressivity as an ambient flow, independently of the orientation of the latent space. In particular, a single scaling layer can learn to absorb the difference in base densities, allowing the flow to squeeze any region of data space into the narrow base density $p_v(v)$ and thus fit the data equally well independent of how the latent variables $u$ and $v$ are aligned with the data manifold. From that perspective it does not seem like PIE is actually a different model than AF. Yet somehow in practice learning dynamics and the inductive bias of the model seem to couple in a way that favor an alignment of the level set $v = 0$ with the data manifold. Understanding these dynamics better would be an interesting research goal.

In many ways, PIE walks and quacks like an ambient flow. In particular, the model density $p_x(x)$ in Eq. (2) generally has support over the full data space $X$, extending beyond the manifold. To sample from this density, one would still draw $u \sim p_u(u)$ and $v \sim p_v(v)$ and apply a transformation $x = f(u, v)$.

However, the labelling of different latent directions as manifold coordinates $u$ and off-the-manifold directions $v$ gives us some new handles. The authors of Reference (21) define a generative mode that samples data only from the learned manifold: one samples $u \sim p_u(u)$ as usually, but fixes $v = 0$, and then applies the transformation $x = f(u, 0)$. This is similar to sampling from the learned manifold for a VAE when the Gaussian mean is used as a deterministic encoder. If the inductive bias of the PIE model successfully leads to an alignment of $u$ with the manifold coordinates, this allows us to sample only from the manifold. Note, however, that the density defined by this sampling procedure is *not* the same as the tractable density $p_x(x)$.[‡] Training a PIE model by maximizing the likelihood

---

[†]To fix the manifold but let the density on it be conditional, one could make $g$ independent of $\theta$ and model $p(u|\theta)$ with a conditional density estimator such as a normalizing flow. Such a partially conditional GAN setup has, to the best of our knowledge, not yet been explored in the literature.

---

[‡]Sampling with $v = 0$ corresponds to the density in Eq. (14), not to the one in Eq. (2). Even when restricted to $x \in \mathcal{M}$, these two densities need not even be proportional to each other. To see this explicitly, we can write the Jacobian of $f$ as $J_f = (J_g, J_\perp)$ in column notation. Then for $x \in \mathcal{M}$ we have

$$p_x(x) = p_u(g^{-1}(x))\, p_v(0) \left| \det \begin{pmatrix} J_g^T J_g & J_g^T J_\perp \\ J_\perp^T J_g & J_\perp^T J_\perp \end{pmatrix} \right|^{-\frac{1}{2}}$$

which is in general not proportional to

$$p_{\mathcal{M}}(x) = p_u(g^{-1}(x)) \left| \det J_g^T J_g \right|^{-\frac{1}{2}}.$$

The discrepancy comes from $p_x(x)$ containing additional factors that describe how the flow "squeezes" and "relaxes" off-the-manifold latent variables around the manifold, but those terms do not play a role for $p_{\mathcal{M}}(x)$. This is the case even if we restrict $f$ to volume-preserving flows.

Brehmer and Cranmer

in Eq. (2) and then sampling from the manifold with $v = 0$ is therefore inconsistent. Finally, note that the hyperparameter $\varepsilon$ allows us to smoothly interpolate between an ambient flow ($\varepsilon = 1$) and "manifolds" ($\varepsilon \ll 1$).

In a conditional version of the original PIE model, both the shape of the manifold as well as the density on it generally depend on the variables $\theta$ being conditioned on. We introduce a new conditional PIE version in which $f$ (and thus the manifold) is independent of $\theta$, while $h(\tilde{u}|\theta)$ and therefore the density are conditional on $\theta$. Training such a model by maximum likelihood leads to a stronger incentive to align the variables $u$ with the manifold coordinates, since in any other alignment the model cannot model the dependence on $\theta$.

**Slice of PIE.** The PIE architecture defines a density $p_x(x)$ over the full data space, and the level set $v = 0$ defines a manifold $\mathcal{M}$. It may therefore be tempting to study the density on $\mathcal{M}$ induced by $p_x(x)$, which is defined as

$$p_{\mathcal{M}'}(x) = \frac{p_x(x)}{\int_{\mathcal{M}} \mathrm{d}x' \, p_x(x')}. \qquad [9]$$

While the normalizing integral in Eq. (9) cannot be computed efficiently, with Eq. (2) we can compute $p_x(x)$ easily enough, so this likelihood is tractable up to an unknown normalizing constant. Depending on the task, this may or may not be sufficient.

The more pressing issue with this model is the generative mode. The density in Eq. (9) is not the same as the density defined by sampling data from the manifold, i.e. drawing $u \sim p_u(u)$ and pushing it into data space with $x = f(u, 0)$. More importantly, we do not know how to sample from Eq. (9) efficiently.[‡]

**Manifold-learning flow ($\mathcal{M}$-flow).** We now introduce the main new algorithm of this paper: the manifold-learning flow or $\mathcal{M}$-flow. It combines the learnable manifold aspect of GANs with the tractable density of flows on manifolds (FOM) without introducing inconsistencies between generative mode and the tractable likelihood. We begin by modeling the relation between the latent space and data space with a diffeomorphism

$$\begin{aligned} f : U \times V &\mapsto X \\ u, v &\to f(u, v), \end{aligned} \qquad [10]$$

just as for an ambient flow or PIE. We define the model manifold $\mathcal{M}$ through the level set

$$\begin{aligned} g : U &\mapsto \mathcal{M} \subset X \\ u &\to g(u) = f(u, 0). \end{aligned} \qquad [11]$$

In practice, we implement this transformation as a zero padding followed by a series of invertible transformations,

$$g = f_k \circ \cdots \circ f_1 \circ \mathrm{Pad}, \qquad [12]$$

where

$$\mathrm{Pad}(u) = \begin{pmatrix} u_0 & \cdots & u_{n-1} & 0 & \cdots & 0 \end{pmatrix}^T \qquad [13]$$

denotes padding a $n$-dimensional vector with $d - n$ zeros and the invertible transformations $f_i$ operate in $d$-dimensional

---

The discrepancy also survives when we consider $p_x(x)/p_v(0)$ in the limit $\varepsilon \to 0$. For a concrete example, think of standard 2D polar coordinates, where $r$ plays the role of $u$ and $\phi$ that of $v$. Let the manifold be given by the line $\phi = 0$. Then $p_x(x)|_{x \in \mathcal{M}} = p_r(r)p_\phi(0)/r$, while $p_{\mathcal{M}}(x) = p_r(r)$.

**Fig. 3.** Sketch of how an $\mathcal{M}$-flow evaluates arbitrary points on or off the learned manifold. On the left side we show the data space $X$ with data samples (grey) and the embedded manifold $\mathcal{M}$ (orange). On the right side the latent space $U \times V$ is shown. In purple we sketch the evaluation of a data point $x$ including its transformation to the latent space, the projection onto the manifold coordinates, and the transformation back to the manifold.

space. Viewed as a map from the latent space $U$ to the data space $X$, the transformation $g$ is injective and (when restricted to its image $\mathcal{M}$) invertible.

Just as for FOM and PIE, we model the base density $p_u(u)$ with an $n$-dimensional flow $h$, which maps $u$ to another latent variable $\tilde{u}$ with an associated tractable base density $p_{\tilde{u}}(\tilde{u})$. There is no need for a base density over the off-the-manifold variables $v$ in this approach. The induced probability density on the manifold is then given by

$$\begin{aligned} p_{\mathcal{M}}(x) &= p_u(g^{-1}(x)) \left| \det[J_g^T(g^{-1}(x)) J_g(g^{-1}(x))] \right|^{-\frac{1}{2}} \\ &= p_{\tilde{u}}(h^{-1}(g^{-1}(x))) \left| \det J_h(h^{-1}(g^{-1}(x))) \right|^{-1} \\ &\quad \times \left| \det[J_g^T(g^{-1}(x)) J_g(g^{-1}(x))] \right|^{-\frac{1}{2}} \quad [14] \end{aligned}$$

This is the same as Eq. (4), except with a learnable transformation $g$ rather than a prescribed, closed-form chart. This model density is defined only on the manifold and normalized to the manifold, $\int_{\mathcal{M}} \mathrm{d}x \, p_{\mathcal{M}}(x) = 1$.

Sampling from an $\mathcal{M}$-flow is straightforward: one draws $\tilde{u} \sim p_{\tilde{u}}(\tilde{u})$ and pushes the latent variable forward to the data space as $u = h(\tilde{u})$ followed by $x = g(u) = f(u, 0)$, leading to data points on the manifold that consistently follow $x \sim p_{\mathcal{M}}(x)$.

As a final ingredient to the $\mathcal{M}$-flow approach, we add a prescription for evaluating arbitrary points $x \in X$, which may be off the manifold. As we illustrate in Figure 3, $g$ maps from a low-dimensional latent space to the data space and is thus essentially a decoder. We define a matching encoder $g^{-1}$ as $f^{-1}$ followed by a projection to the $u$ component:

$$\begin{aligned} g^{-1} : X &\mapsto U \\ x &\to g^{-1}(x) = \mathrm{Proj}(f^{-1}(x)) \qquad [15] \end{aligned}$$

with $\mathrm{Proj}(u, v) = u$. This extends the inverse of $g$ (which is so far only defined for $x \in \mathcal{M}$) to the whole data space $X$. Similar to an autoencoder, combining $g$ and $g^{-1}$ allows us to calculate a reconstruction error

$$\|x - x'\| = \|x - g(g^{-1}(x))\|, \qquad [16]$$

which is zero if and only if $x \in \mathcal{M}$. Unlike for standard autoencoders, the encoder and decoder are exact inverses of each other as long as points on the manifold are studied.

For an arbitrary $x \in X$, an $\mathcal{M}$-flow thus lets us compute three quantities:

- The projection onto the manifold $x' = g(g^{-1}(x))$, which may be used as a denoised version of the input.

- The reconstruction error $\|x - x'\|$, which will be important for training, but may also be useful for anomaly detection or out-of-distribution detection.

- The likelihood on the manifold after the projection, $p_{\mathcal{M}}(x')$.

In this way, $\mathcal{M}$-flows separate the distance from the data manifold and the density on the manifold—two concepts that easily get conflated in an ambient flow. $\mathcal{M}$-flows thus embrace ideas of energy-based models for dealing with off-the-manifold issues, but still have a tractable, exact likelihood on the learned data manifold. Figure 3 summarizes how an $\mathcal{M}$-flow model evaluates a data point $x \in X$ by transforming to the latent space, projecting onto the manifold (where the density is evaluated), and transforming back to data space (where the reconstruction error is calculated).

**Manifold-learning flows with separate encoder ($\mathcal{M}_e$-flow).** Finally, we introduce a variant of the $\mathcal{M}$-flow model where instead of using the inverse $f^{-1}$ followed by a projection as an encoder, we encode the data with a separate function

$$
\begin{aligned}
e : X &\mapsto U \\
x &\to e(x) \,.
\end{aligned} \tag{17}
$$

This encoder is not restricted to be invertible or to have a tractable Jacobian, potentially increasing the expressiveness of the network. Just as in the $\mathcal{M}$-flow approach, for a given data point $x \in X$, this $\mathcal{M}_e$-flow model returns a projected point onto the learned manifold $g(e(x))$, a reconstruction error $\|x - g(e(x))\|$, and the likelihood on the manifold evaluated after the projection

$$
p_{\mathcal{M}}(x) = p_u(e(x)) \ \left| \det[J_g^T(e(x)) J_g(e(x))] \right|^{-\frac{1}{2}} \,. \tag{18}
$$

The added expressivity of this encoder comes at the price of potential inconsistencies between encoder and decoder, which the training procedure will have to try to penalize, exactly as for a standard autoencoders and similar to VAEs.

**Probabilistic autoencoder (PAE).** How important is the invertibility of the transformation $f$ (and therefore $g$) in the $\mathcal{M}$-flow and $\mathcal{M}_e$-flow models? If we take the $\mathcal{M}_e$-flow model and replace the invertible transformation $g$ of Eq. Eq. (11) with a decoder

$$
\begin{aligned}
g : U &\mapsto X \\
u &\to g(u)
\end{aligned} \tag{19}
$$

that is not required to be invertible, we arrive at an autoencoder model in which the latent space is modeled with a flow. This setup has recently been called probabilistic autoencoder (PAE) (22). While relaxing the requirement of invertibility may make this generative model more expressive, it also loses the tractable density of the model.

**D. Manifolds with unknown dimensionality or nontrivial topology.** So far we have made two key assumptions to simplify the learning problem: that we know the manifold dimensionality $n$ and that the manifold is topologically equivalent to $\mathbb{R}^n$ (in particular that it can be mapped by a single chart).

The algorithms presented above can be extended to the more general case where these assumptions are relaxed.

If the dimension of the manifold is not known, a brute-force solution would be to scan over values of $n$ and train algorithms for each value. A common metric for flow-based models is the model log likelihood evaluated on a number of test samples, but that criterion is not admissible in this context since the space of the data (and the units of the likelihood) are different for different values of $n$. However, we can compare models with different manifold dimensionality based on the reconstruction error, as well as on downstream tasks such as evaluating the quality of generated samples or the performance on inference tasks. A drop in performance is expected when the model manifold becomes smaller than the true manifold dimension.

Alternatively, for the PIE algorithm one could use trainable values of the base density variance $\varepsilon$ along each latent direction, with suitable regularization favoring values close to 0 or 1. In this way the model can learn the manifold dimensionality directly from the training data.

If the manifold consists of multiple disjoint pieces, potentially with different dimensionality, a mixture model with separate transformations from latent space to data space may work. It remains to be seen if such a model is easy to train. See Reference (16) for a discussion of such issues.

## 3. Efficient training and evaluation

Having defined the $\mathcal{M}$-flow model, we will now turn to the question is how to train it. Most flow-based generative models are trained by maximum likelihood, with architectures commonly designed with the goal of making the likelihood in Eq. (2) efficient to evaluate. For implicit generative models that is not available: GANs are trained adversarially, for instance pitted against a discriminator or using an optimal transport (OT) metric, while VAEs are commonly trained on a lower bound for the marginal likelihood (the ELBO). We will draw on all of these approaches, beginning with a discussion of two challenges of likelihood-based training for $\mathcal{M}$-flows in Section 3.A. We discuss a number of more promising training strategies in Section 3.B, before commenting on steps to also make the evaluation of the likelihood more efficient in Section 3.C.

### A. Maximum likelihood is not enough.

**A subtlety in the naive interpretation of the density.** Since the $\mathcal{M}$-flow model has a tractable density, maximum likelihood is an obvious candidate for a training objective. However, the situation is more subtle as the $\mathcal{M}$-flow model describes the density *after projecting onto the learned manifold*. The definition of the data variable in the likelihood hence depends on the weights $\phi_f$ of the manifold-defining transformation $f$, and a comparison of naive likelihood values between different configurations of $\phi_f$ is meaningless. Instead of thinking of a likelihood function $p(x|\phi_f, \phi_h)$, where $\phi_h$ are the weights of the transformation $h$, it is instructive to think of a family of likelihood functions $p_{\phi_f}(x|\phi_h)$ parameterized by the different $\phi_f$.

Training $\mathcal{M}$-flows by simply maximizing the naive likelihood $p(x|\phi_f, \phi_h)$ is therefore not meaningful, does not incentivize the network to learn the right shape of the manifold, and probably will not converge to the true model. As an extreme example, consider a model manifold that is perpendicular to

**(a)** Setup. The model manifold is a straight line in 2D Euclidean space that passes through the origin and is rotated with respect to the $x$-axis by an angle $\alpha$. On this line, the density is a Gaussian with mean at the origin, its standard deviation is a model parameter $\sigma$. The training data (black dots) are generated with $\alpha^* = \pi/2$ and $\sigma^* = 1$.

**(b)** Loss functions. **Top left**: naive log likelihood as a function of the model parameters $\alpha$ and $\sigma$. When fixing the manifold to $\alpha = \pi/2$, the true value $\sigma = 1$ (black star) maximizes the naive likelihood. However, when varying both parameters, the likelihood can be larger for the pathological configuration $\alpha \to 0$ and $\sigma \to 0$. **Top right**: reconstruction error when projecting to the model manifold, which is minimized by the true configuration $\alpha = \pi/2$. **Bottom left**: combined loss given by the reconstruction error minus a small factor times the naive log likelihood—the true configuration is a local minimum, but the global minimum for $\alpha \to 0$ and $\sigma \to 0$ persists. **Bottom right**: Log likelihood after subtracting the maximum log likelihood for each value of $\alpha$.

**Fig. 4.** Toy example showing that maximum naive likelihood is not a suitable training objective for manifold-learning flows.

the true data manifold. Since this configuration allows the $\mathcal{M}$-flow to project all points to a region of very high density on the model manifold, this pathological configuration may lead to a very high naive likelihood value.

We demonstrate this issue in Figure 4 in a simple toy problem. The ambient data space is two-dimensional, the model manifold consists of a line through the origin with variable angle $\phi_f = \alpha$ such that $\alpha = 0$ corresponds to a manifold aligned with the $x$-axis and $\alpha = \pi/2$ to a manifold aligned with the $y$-axis. On this line we consider a one-dimensional Gaussian probability density with mean at the origin and standard deviation $\phi_h = \sigma$. Training samples are generated from $\alpha^* = \pi/2$ and $\sigma^* = 1$. The setup is sketched in Figure 4a. In the top left panel of Figure 4b we show how the naive likelihood of this model over the training data depends on the parameters $\alpha$ and $\sigma$. When fixing the manifold to the true value $\alpha = \pi/2$, the correct standard deviation $\sigma = 1$ indeed maximizes the naive likelihood. However, the model can achieve an even higher naive likelihood for $\alpha \to 0$ and $\sigma \to 0$, representing a manifold that is orthogonal to the true one and projects all data points to a region of extremely high density on the manifold. In this limit the likelihood is in fact unbounded from above. Clearly, maximizing the naive $p(x|\alpha, \sigma)$ alone is not very good at incentivizing the model to learn the correct manifold.

To address this, we can add a second training objective that is responsible for learning the manifold. A suitable candidate is the reconstruction error $\|x - x'\|$ discussed in the previous

section. The top right panel in Figure 4b shows the mean reconstruction error as a function of the model parameters, which is indeed minimal for the true configuration.

One way to combine the two metrics is training on a combined loss that sums reconstruction error and negative naive log likelihood, with hyperparameters $\lambda$ weighting the two terms. This helps, but does not really solve the problem. In our toy example we show such a combined loss in the bottom left of Figure 4b. While the correct configuration is a local minimum of this loss, the wrong minimum at $\alpha \to 0$ and $\sigma \to 0$ still exists and leads to a lower (and unbounded from below) loss. In general the correct solution might not even be a local minimum of such a combined loss function. When training the model parameters by minimizing this combined loss, the gradient flow may take the model to the correct solution or a pathological configuration, depending on the initialization and the choice of hyperparameters.

A better strategy is to separate the model parameters that define the manifold from the ones that only describe the density on them. In the $\mathcal{M}$-flow setup in the previous section, the parameters of the transformation $f$ (or $g$) make up the first class, while the parameters of $p_u$ (or $h$) are in the second; in the toy example in Figure 4 $\alpha$ fixes the manifold and $\sigma$ the density on it. We can then update the manifold parameters based on only the reconstruction error and update the density weights based on only the log likelihood. In Figure 4b this corresponds to horizontal steps in the top right panel and vertical steps in the bottom right panel, where we show the log likelihood

normalized to the maximum likelihood estimator (MLE) for each value of $\alpha$. Such a training procedure is not prone to the gradient flow leading the model to a pathological configuration. In the limit of infinite capacity, sufficient training data, and succesful optimization, it will correctly learn both the manifold and the density on it.

**Evaluating the likelihood can be expensive.** The second challenge is the computational efficiency of evaluating the $\mathcal{M}$-flow density in Eq. (14). While this quantity is tractable, it cannot be computed as cheaply as the ambient flow density of Eq. (2). The underlying reason is that since the Jacobian $J_g$ is not square, it is not obvious how the determinant $\det J_g^T J_g$ can be decomposed further. In particular, when the map consists of multiple functions as given in Eq. (12), the Jacobian is given by a product of individual Jacobians, $J_g = (\prod_i J_i) J_{\text{Pad}}$. While the $J_i$ are invertible $d \times d$ matrices, the Jacobian that represents the zero-padding is a rectangular matrix that consists of a $n \times n$ identity matrix padded with zeros, which leaves us with the following Jacobian to calculate:

$$\log p_\mathcal{M} = \cdots - \frac{1}{2} \log \det \left[ (\mathbb{1} \ 0) J_1^T \ldots J_k^T J_k \ldots J_1 \begin{pmatrix} \mathbb{1} \\ 0 \end{pmatrix} \right]. \quad [20]$$

This determinant *can* be computed explicitly. However, when we compose an $\mathcal{M}$-flow out of invertible transformations that have been designed for standard flows—coupling layers with invertible elementwise transformations, autoregressive transformations, permutations, or invertible linear transformations—evaluating this $\mathcal{M}$-flow density requires the computation of all entries of the Jacobians of the individual transformation. This is a much larger computational effort than in the case of standard flows, where the overall log determinant can be split into a sum over the log determinants of each layer, which in turn can usually be written down as a single number without having to compute all elements of a Jacobian first.

While the computational cost of evaluating Eq. (20) is often reasonable for the evaluation of a limited number of test samples, it can be prohibitively expensive during training, which often requires many more evaluations. Since the computational cost grows with increasing data dimensionality $d$, training by maximizing $\log p_\mathcal{M}$ does not scale to high-dimensional problems.

Fortunately, gradient updates do not always require computing the full likelihood of the model. In particular, consider the training procedure introduced in the previous section, where we update the parameters of $f$ by minimizing the reconstruction error and update the parameters of $h$ (and thus of $p_u$) by maximizing the log likelihood. The manifold update phase does not require computing the log likelihood at all. For the density update, the loss functional $L[h]$ is given by

$$
\begin{aligned}
L[h] &= -\frac{1}{N} \sum_x \log p_\mathcal{M}(x) \\
&= -\frac{1}{N} \sum_x \Big( \log p_{\tilde{u}}(h^{-1}(u)) - \log \det J_h(h^{-1}(u)) \\
&\qquad\qquad - \frac{1}{2} \log \det [J_g^T(u) J_g(u)] \Big),
\end{aligned}
\quad [21]
$$

with $u = g^{-1}(x)$. However, the last term (which is slow to evaluate) does not depend on the parameters of $h$ and does not contribute to the gradient updates in this phase! We can

therefore just as well train the parameters of $h$ by minimizing only the first two terms, which can be evaluated very efficiently.

## B. Training strategies.

**Simultaneous manifold and likelihood training (S).** For completeness we include here the simultaneous optimization of the parameters of the manifold-defining transformation $f$ and the parameters of the density-defining transformation $h$ on a combined loss summing negative naive log likelihood and reconstruction error,

$$L_\text{S}[g, h] = \frac{1}{N} \sum_x \left( -\log p_\mathcal{M}(x) + \lambda \|x - g^{-1}(g(x))\| \right), \quad [22]$$

where $\lambda$ is a hyperparameter.

Following the discussion in Section A, we do not expect this algorithm to perform very well. First, as demonstrated in the toy example in Figure 4 there is a risk of pathological models with poor manifold quality and poor density estimation for which this loss is very small, potentially even lower than for the true model. Which configuration the model ends up in may critically depend on the initialization and the learning dynamics. Second, evaluating this loss can be computationally expensive, especially for high-dimensional problems. Nevertheless, we include this in our experiments on low-dimensional data for comparison.

In order to ameliorate both the potential instability of this training objective and to speed up the training, we add a pre-training and a post-training phase. In the pre-training, the model is trained by minimizing the reconstruction error only, hopefully pushing the weights of $f$ into the basin of attraction around the true model configuration, before the main training phase begins. In the post-training phase, the parameters of $f$ are fixed and only the parameters of $h$ are updated by minimizing only the relevant terms in the loss.

**Separate manifold and density training (M/D)** As discussed above, we expect a both faster and more robust training when separating manifold and density updates, splitting the training into two phases:

**Manifold phase:** Update only the parameters of $f$ (and thus also $g$, which is defined as a level set of $f$) by minimizing the squared reconstruction error from the projection onto the manifold,

$$L_\text{manifold}^\text{M/D}[g] = \frac{1}{b} \sum_x \|x - g(g^{-1}(x))\|_2^2 \quad [23]$$

with batch size $b$. For the $\mathcal{M}_e$-flow model, the parameters of the encoder $e$ are also updated during this phase.

**Density phase:** Update only the parameters of $h$ (which define the coordinate density $p_u$) by minimizing the negative log likelihood

$$L_\text{density}^\text{M/D}[h] = -\frac{1}{b} \sum_x \log p_u(g^{-1}(x)). \quad [24]$$

An important choice is how these two phases are scheduled. The most straightforward strategy is a *sequential* training, in which the manifold-defining transformation $f$ is learned first, followed by the density-defining transformation $h$. We also

Brehmer and Cranmer

---

**Algorithm 1** Alternating manifold / density (M/D) training for manifold-learning flows. Instead of alternating between manifold and density phases as shown here, one can employ a sequential version in which the manifold is first trained until convergence, followed by a training phase focused on the density. For simplicity we show a version based on stochastic gradient descent with a constant learning rate, though the algorithm can trivially be extended to other optimizers and learning rate schedules (which may be different for the two phases).

---

**Require:** $\alpha$, the learning rate. $\lambda_m$, $\lambda_d$, factors weighting terms in the loss functions. $b_m$ and $b_d$, the batch sizes. $N_m$ and $N_d$, the number of batches per training phase.
**Require:** $\phi_f^0$ and $\phi_h^0$, initial weights of the $\mathcal{M}$-flow transformations $f$ and $h$.
   $\phi_f \leftarrow \phi_f^0$
   $\phi_h \leftarrow \phi_h^0$
   **while** $\phi_f$ has not converged or $\phi_h$ has not converged **do**

      **for** $t = 1, \ldots, N_m$ **do**                                                                 ▷ Manifold phase
         Sample $\{x_i\}_{i=1}^{b_m} \sim p^*(x)$                                         ▷ Sample training data
         **for** $i = 1, \ldots, b_m$ **do**
            $u_i, v_i \leftarrow f^{-1}(x_i)$                                    ▷ Transform to latent space
            $x_i' \leftarrow f(u_i, 0)$           ▷ Project to manifold, transform back to data space
         $L_{\mathrm{manifold}} \leftarrow \lambda_m / b_m \sum_i \|x_i - x_i'\|_2^2$             ▷ Compute reconstruction error
         $\phi_f \leftarrow \phi_f - \alpha \nabla_{\phi_f} L_{\mathrm{manifold}}$                  ▷ Update manifold weights

      **for** $t = 1, \ldots, N_d$ **do**                                                                   ▷ Density phase
         Sample $\{x_i\}_{i=1}^{b_d} \sim p^*(x)$                                        ▷ Sample training data
         **for** $i = 1, \ldots, b_d$ **do**
            $u_i, v_i \leftarrow f^{-1}(x_i)$                                    ▷ Transform to latent space
            $\tilde{u}_i' \leftarrow h^{-1}(u_i)$       ▷ Project to manifold, transform to coordinate base space
         $L_{\mathrm{density}} \leftarrow -\lambda_d / b_d \sum_i [\log p_{\tilde{u}}(\tilde{u}_i) - \log \det J_h(\tilde{u}_i)]$     ▷ Compute log likelihood
         $\phi_h \leftarrow \phi_h - \alpha \nabla_{\phi_h} L_{\mathrm{density}}$                   ▷ Update density weights

---

experiment with an *alternating* scheme, where we switch between the two phases after a fixed number of gradient updates. The algorithm is described in more detail in Algorithm 1.

To study the convergence of the model under this training schedule we can separately study whether it reaches the correct manifold shape (defined by $f$) and whether it correctly models the true density on the manifold (defined by $f$ and $h$). First, the ability of $\mathcal{M}$-flows to converge to the correct *manifold* is essentially the same as the considerations for autoencoders, with an additional architectural requirement of invertibility. For data on a manifold that can be described by a single chart with the true latent space dimensionality $n$ (which here we assume to be known), this does not pose a restriction; this is related to the fact that all submanifolds that satisfy modest regularity conditions can be expressed as level sets of bijections. Second, if $f$ has converged and learned the manifold, then learning the *density* on the manifold is a $n$-dimensional density estimation task. By implementing $h$ as a flow that is a universal density approximator, we ensure that the $\mathcal{M}$-flow model can express any density on the manifold (up to some regularity assumptions).

**Adversarial training (OT).** Another option is to train manifold-learning flows adversarially, similar to GANs or Flow-GANs (23). The loss function is then a distance metric between samples generated from the $\mathcal{M}$-flow model and a batch of training samples. Such a distance metric can for instance be based on the output of a discriminator that is trained simultaneously, or an integral probability metric such as the Wasserstein distance. We use unbiased Sinkhorn divergences, a tractable but positive definite approximation of Wasserstein divergences (24). In this training scheme, which we label OT, we iterate over the data in mini-batches $\{x\}$, generate equally sized batches of samples $\{x_{\mathrm{gen}}\}$ from the manifold-learning flow, and update the gradients based on the loss

$$L_{\mathrm{OT}}[g,h] = S_\varepsilon(\{x\}, \{x_{\mathrm{gen}}\})\Big|_{\{x_{\mathrm{gen}}\} \sim p_{\mathcal{M}}(x)}. \qquad [25]$$

Here the Sinkhorn divergence is defined as

$$S_\varepsilon(\{x\}, \{x_{\mathrm{gen}}\}) = OT_\varepsilon(\{x\}, \{x_{\mathrm{gen}}\}) - \frac{1}{2} OT_\varepsilon(\{x\}, \{x\})$$
$$- \frac{1}{2} OT_\varepsilon(\{x_{\mathrm{gen}}\}, \{x_{\mathrm{gen}}\}) \quad [26]$$

with entropy-regularized optimal transport loss $OT_\varepsilon$. $S_\varepsilon(\{x\}, \{x_{\mathrm{gen}}\})$ interpolates between Wasserstein distance (for $\varepsilon \to 0$) and energy distance (for $\varepsilon \to \infty$). See Reference (24) for a detailed explanation.

**Alternating adversarial and likelihood training (OT/D).** We can combine this adversarial training with likelihood-based phases for the base density $p_u(u)$ into an alternating algorithm. It is essentially the same as the M/D algorithm described in Algorithm 1, except that in the first phase we draw samples from the model as well and optimize the parameters of both $f$ and $h$ by minimizing the loss in Eq. (25).

**Geometric implicit regularization.** Given a set of data points $\{x\}$, it is possible to train a neural network $d(x)$ that maps the data space to $\mathbb{R}$ to learn a signed distance function from the data manifold. The level set $d(x) = 0$ then corresponds to the manifold. Reference (25) proposes to achieve this goal by minimizing

$$L_{\mathrm{reg}}[d] = \frac{1}{N} \sum_x |d(x)| + \lambda \mathbb{E}_{x'}(\|\nabla_{x'} d(x')\| - 1)^2, \qquad [27]$$

combining a term that favors the network to be zero on the data with an "Eikonal" term that encourages the gradients $\nabla_x d$ to be of unit norm everywhere, weighted by a hyperparameter $\lambda$. The expectation is taken with respect to some probability distribution over $X$.

This ansatz can be applied to manifold-learning flows. One approach would be to add a term like Eq. (27) for each component of the off-the-manifold latent variables $v$ to the existing loss functions,

$$L_{\text{combined}}[g, h] = L[g, h] + \alpha \sum_i L_{\text{reg}}[v_i(x)]. \qquad [28]$$

Computing this regularization term then requires the evaluation of the Jacobian $\partial v / \partial x$, which is plagued by the same computational inefficiency that we discussed for $J_g \sim \partial x / \partial u$ before. Nevertheless, the authors of Reference 27 report learned manifolds of a very high quality even for few training samples, and the computational expense may well be worth it. We leave an exploration of this idea for future work.

**C. Likelihood evaluation.** Above we discussed training strategies that avoid a computation of the expensive terms in the likelihood. Even with such an efficient training, the model likelihood often needs to be evaluated at test time, although typically not quite as often. Here we collect ideas for how to improve the efficiency of the likelihood evaluation.

**Exact likelihood.** While the model likelihood in Eq. (14) is tractable, evaluating it for typical flow transformations can be somewhat slow. The cost of this evaluation increases with the dimension of the data space as well as with the complexity of the network architecture. In our experiments we found that this cost is not the limiting factor when evaluating low- to medium-dimensional data spaces, even in the context of inference problems that require many repeated evaluations of the likelihood. In this work we thus restricted ourselves to exact likelihood evaluations and did not study the methods described in the following further.

**Efficient exact inference on model parameters.** A common downstream task is inference on some model parameters $\theta$ that the model density $p(x|\theta)$ depends on. Often the data manifold is independent of these parameters and only the density on the manifold depend on them. We argued above that we can incorporate this structure into the $\mathcal{M}$-flow setup by making the manifold-defining transformation $f$ independent of $\theta$ and only letting the density-defining transformation $h$ be conditional on $\theta$. This setup may improve the performance, as the structure is built into the architecture and does not have to be learned.

On top of that, such a setup allows for exact, efficient inference on $\theta$. Consider the likelihood ratio between two different parameter points $\theta_0$ and $\theta_1$,

$$\frac{p(x|\theta_0)}{p(x|\theta_1)} = \frac{p_{\tilde{u}}(h^{-1}(u; \theta_0))}{p_{\tilde{u}}(h^{-1}(u; \theta_1))} \frac{\left|\det J_h(h^{-1}(u; \theta_0))\right|^{-1}}{\left|\det J_h(h^{-1}(u; \theta_1))\right|^{-1}}$$

$$\times \frac{\left|\det[J_g^T(u) J_g(u)]\right|^{-\frac{1}{2}}}{\left|\det[J_g^T(u) J_g(u)]\right|^{-\frac{1}{2}}}, \quad [29]$$

where $u = g^{-1}(x)$ is independent of $\theta$. The expensive Jacobian terms cancel in the ratio! Similarly, they drop out of the

acceptance proabbility of MCMC samplers. Inference on model parameters $\theta$ thus does not require the computation of the slow terms in the likelihood function, provided that the data manifold is independent of $\theta$.

**Approximate likelihood.** The likelihood in Eq. (14) can be computed approximately, for instance with the methods proposed in References (19, 26, 27). Instead of computing the full matrix $J_g^T J_g$, these methods just require calculating a number of matrix-vector products $J_g^T J_g u$ with randomly sampled vectors $u$, which can be cheaper. Whether the gains in speed are worth the loss in precision from the approximation remains to be seen; we leave a test of this idea for future work.

**Approximate lower bound on the likelihood.** The authors of Reference (10) derive a lower bound on the likelihood in Eq. (14). While the lower bound itself is computationally expensive, they derive a stochastic estimator for it that can be computed efficiently. Again we leave an exploration of the idea for our model for future work.

**Regression on the Jacobian determinant.** The cost of evaluating the Jacobian determinant in Eq. (14) can be amortized by evaluating this Jacobian factor for a number of representative data points first, and then regressing on the function $j(u) = \log \det[J_g^T(u) J_g(u)]$. Afterwards, the $\mathcal{M}$-flow likelihood can be evaluated at any point efficiently. We leave an investigation of this idea for future work.

**Optimized architectures.** The characterization of the evaluation of the Jacobian in Eq. (14) as computationally expensive depends on the architecture of the transformation $f$. In this work we only consider zero-padding followed by typical diffeomorphic transformations like coupling layers with invertible elementwise transformations or permutations; these transformations have evolved over many years of research with the design goal of efficient standard flow densities in mind. It is well possible that a similar amount of research will unveil a class of transformations for which the terms in Eq. (14) can be computed efficiently without limiting their expressiveness. We hope that this paper can instigate research into such transformations.

## 4. Experiments

We will now demonstrate manifold-learning flows in four experiments. We begin with two pedagogical examples, before analyzing the Lorenz attractor in Section 4.C, a real-life particle physics dataset in Section 4.D, and finally image datasets in Sections 4.E and 4.F.

A common metric for flow-based models is the model log likelihood evaluated on a test set, but such a comparison is not meaningful in our context. Since the $\mathcal{M}$-flow variants evaluate the likelihood after projecting to the learned manifolds, the data variable in the likelihood is different for every model and the likelihoods of different models may not even have the same units. Instead, we analyze the performance through the generative mode, evaluating the quality of samples generated from the models with different metrics depending on the dataset. In addition, we use the model likelihood for inference tasks and gauge the quality of the resulting posterior.

**A. Gaussian on a circle.** First, we want to illustrate the different flow models in a simple toy example. Data is generated

**Fig. 5.** Learning a Gaussian density on a circle. **Top left**: true density of the data-generating process. **Top middle and top right**: 2D density learned by a standard ambient flow (AF) and a PIE). **Bottom**: manifold and density learned by a manifold flow with specified true manifold (FOM), a manifold-learning flow ($\mathcal{M}$-flow (M/D)), and a manifold-learning flow that was only trained on the reconstruction error ($\mathcal{M}$-flow (AE)). To highlight the differences, we use simple, less expressive architectures (see text).

on a unit circle in two-dimensional space, where the usual polar angle $\phi$ is drawn from a Gaussian density with mean $\pi/2$ and standard deviation $\pi/4$. To represent a slightly noisy true manifold, the radial coordinate is not exactly set to 1, but drawn from a Gaussian density with mean 1 and standard deviation 0.01. As training data, we generate $10^4$ points in this way.

To highlight the difference between the different models, we purposefully limit the expressivity of the flows by using simple affine coupling layers interspersed with random permutations of the latent variables. For the ambient flow we use ten affine coupling layers, while for PIE and the $\mathcal{M}$-flow variants we restrict $f$ to five such layers and model $p_u(u)$ with a Gaussian with learnable mean and variance. We also consider a FOM model, using the known parameterization of the unit circle to model the manifold and a Gaussian with learnable mean and variance for $p_u(u)$. Finally, for demonstration purposes we also include an $\mathcal{M}$-flow model that is only trained on reconstruction error, essentially an invertible auto-encoder, in this study and label it $\mathcal{M}$-flow (AE). In all cases, we limit the training to 120 epochs.

Figure 5 shows the true density of the data-generating process (top left) as well as the learned densities from different models (other panels). The standard flow (AF, top middle) learns a smeared-out version of the true density, with a substantial amount of probability mass away from the true manifold. Note that the AF results become much sharper when we train until convergence or switch to a state-of-the-art architecture, as we have tested with rational-quadratic neural

spline flows (28). The PIE model (top right) also learns a smeared-out version, but its inductive bias leads to a sharper version than the AF model. We also show the manifold represented by the level set $v = 0$ in the PIE model as a dotted black line, it is not in particularly good agreement with the true manifold.

In the bottom panels we show some algorithms with a model density restricted to the manifold, the black space in the figures thus shows the off-the-manifold region which are outside the support of the model. Note that this different support also means that the likelihood values between the top and bottom panels cannot be directly compared. The FOM model (bottom left), which requires knowledge of the manifold, perfectly captures both the shape of the manifold and the density on it. Our new $\mathcal{M}$-flow (M/D) algorithm (bottom middle) also parameterizes the density only on the manifold, but now the manifold is learned from data; we see both good manifold quality and good density estimation in the upper half of the circle, where most of the training data lie. In the lower part, where the density was too small to sample enough training data, the learned manifold departs from the true one. Finally, in the bottom right panel we show that training an $\mathcal{M}$-flow model just on reconstruction error can lead to a good approximation of the manifold (where there is training data), but, of course, does not produce a reasonable density on this manifold.

**Fig. 6.** Mixture model on a polynomial surface. **Top**: the true data manifold as well as the manifolds learned by the PIE, $\mathcal{M}$-flow (M/D), and $\mathcal{M}$-flow (OT) models. The color shows the log likelihood for $\theta = 0$ (bright yellow represents a high density, dark blue a low density). In order to increase the clarity of the PIE panel we have removed parts of that manifold which "fold" above and below the shown part. **Bottom**: ground truth and $\mathcal{M}$-flow (M/D) manifold for $\theta = -1$ and $\theta = 1$.

**B. Mixture model on a polynomial surface.** Next, we consider a two-dimensional manifold embedded in $\mathbb{R}^3$ defined by

$$x = R \begin{pmatrix} z_0 \\ z_1 \\ f(z) \end{pmatrix} \quad \text{with} \quad f(z) = \exp(-0.1\|z\|) \sum_{i+j \leq 6} a_{ij} z_0^i z_1^j .$$

[30]

Here $R \in \mathrm{SO}(3)$ is a three-dimensional rotation matrix, $z = (z_0, z_1)$ is a vector of two latent variables that parameterize the manifold, $a_{ij}$ are the coefficients of a polynomial, and $N$ is the maximal power in the series. We choose a fixed value for $R$ and $a_{ij}$ for these experiments by a single random sampling from the Haar measure and normal distributions, the values of these parameters are given in the appendix.

We define a conditional probability density on the latent variables as

$$p(z|\theta) = 0.6\mathcal{N}\left(z \middle| \begin{pmatrix} 1 \\ -1 \end{pmatrix}, 2^2 \cdot \mathbb{1} \right)$$
$$+ 0.4\mathcal{N}\left(z \middle| \begin{pmatrix} -1 \\ 1 \end{pmatrix}, (0.6 + 0.4\theta)^2 \cdot \mathbb{1} \right), \quad [31]$$

which together with the chart in Eq. (30) defines a probability density on the manifold. The dominant component of this mixture model is thus a normal distribution with a large covariance that is independent of the parameter $\theta$, while only the covariance of the smaller component depends on the parameter $\theta$, which is restricted to the range $-1 \leq \theta \leq 1$.

We train several manifold-learning flow variants on $10^5$ training samples and compare to AF, PIE, and PAE baselines. In addition to the original PIE model, in which the manifold-defining transformation $f$ is conditional on the parameters $\theta$, we also include a model where this transformation is independent of $\theta$ and only the density on $\mathcal{M}$ is conditional on $\theta$. In

all cases we use rational-quadratic neural spline flows with ten coupling layers interspersed with random permutations of the features. The setup is described in detail in the appendix.

We visualize the true data manifold and the estimated manifolds from a few $\mathcal{M}$-flow and PIE models in Figure 6. In the top panels we compare the ground truth and three trained models conditional on $\theta = 0$, in the bottom panels we show how the ground truth and the $\mathcal{M}$-flow (M/D) model

**Fig. 7.** Mixture model on a polynomial surface. On the two-dimensional slice defined by $x_0 = 0$, we show cross sections through the true data manifold as well as the manifolds learned by the PIE and $\mathcal{M}$-flow (M/D) models. The color shows the log likelihood for $\theta = 0$ (bright yellow represents a high density, dark blue a low density, and black regions off the support of the model).

change for $\theta = \pm 1$. The manifold defined by $v = 0$ in the PIE model is clearly not a good approximation of the true manifold—these directions are only partially aligned with the true data manifold, and the surface defined in this way does not extend near a large part of the true data manifold at all. The $\mathcal{M}$-flow (OT) model gets some of the features of the manifold and density right, but does not perform very well in regions of low density. The results that most closely resemble the true model come from the $\mathcal{M}$-flow (M/D) model: not only are the learned manifold and the density of the manifold very similar to the ground truth, but the model accurately captures the dependency of the likelihood on the model parameter $\theta$.

Figure 7 shows a cross section of the ground truth, AF, and $\mathcal{M}$-flow models. While the AF density is sharply peaked around the true manifold and most of its probability mass is very close to it, it nevertheless has support off the manifold, especially in regions of low density. The $\mathcal{M}$-flow model, on the other hand, exactly learns a two-dimensional manifold.

In Table 2 we evaluate the performance of the models on four metrics:

- We compare the quality of samples generated from the flows by calculating the mean distance from the true data manifold using Eq. (30), as described in the appendix.

- For all models except the AF we calculate the mean reconstruction error when projecting test samples onto the learned manifold.

- We use the flow models for approximate inference on the parameter $\theta$. We generate posterior samples with an MCMC sampler, using the likelihood of the different flow models in lieu of the true simulator density. The results are compared to posterior samples based on the true simulator likelihood. We summarize the similarity with the maximum mean discrepancy (MMD) of the posterior samples based on a Gaussian kernel (29).

- Finally, we evaluate out-of-distribution (OOD) detection. For each model, we compare the distribution of log likelihood and reconstruction error between a normal test sample based on Eq. (31) and an OOD sample. The latter is based on the same density as the original model plus

Gaussian noise with zero mean and standard deviation of 0.1 on all three features, pushing it off the data manifold of the regular training and test samples. We report the area under the curve (AUC), giving the larger number when both discrimination based on model likelihood and reconstruction error is available.

For each metric, we report the median based on five runs with independent training samples and weight initializations.

In all metrics except the out-of-distribution detection, manifold-learning flows provide the best results. In particular, samples generated from the $\mathcal{M}$-flow (M/D) and $\mathcal{M}_e$-flow (M/D) models are closer to the true data manifold than those from the AF and PIE models. The $\mathcal{M}$-flow (M/D) and $\mathcal{M}_e$-flow (M/D) models also learned a higher-quality manifold than PIE, as measured by the reconstruction error when projecting test samples the learned manifold. In both of these metrics, they are competitive with PAE, which does not have a tractable likelihood, demonstrating that the restriction to an invertible decoder does not limit the flexibility of the models too much.

When it comes to inference on $\theta$, the $\mathcal{M}$-flow variants clearly outperform the AF and PIE baselines (we cannot compare to PAE due to its intractable likelihood). For out-of-distribution detection, the reconstruction error returned by these manifold-learning flows is not quite as good as the AF log likelihood and the PAE reconstriction error. The other training algorithms all have their shortcomings: $\mathcal{M}$-flow (S) training is not only slower, but also leads to slightly worse results, and the optimal transport variants $\mathcal{M}$-flow (OT) and $\mathcal{M}$-flow (OT/D) do not perform well on any metric, perhaps signalling the need for a more thorough tuning of hyperparameters.

**C. Lorenz attractor.** The Lorenz system (30) is a three-dimensional, non-linear, deterministic system originally developed by Edward Lorenz as a simplified model for atmospheric convection. A three-dimensional vector $x$ evolves with time

| Model (algorithm) | Mean distance from manifold | Mean reconstruction error | Posterior MMD | Out-of-distribution AUC |
|---|---|---|---|---|
| AF | 0.005 | – | 0.071 | **0.990** |
| PIE (original) | 0.035 | 1.278 | 0.131 | 0.933 |
| PIE (unconditional manifold) | 0.006 | 1.253 | 0.075 | 0.972 |
| PAE | **0.002** | **0.002** | – | **0.990** |
| $\mathcal{M}$-flow (S) | 0.006 | 0.011 | **0.026** | 0.974 |
| $\mathcal{M}$-flow (alternating M/D) | **0.002** | **0.003** | **0.020** | **0.986** |
| $\mathcal{M}$-flow (sequential M/D) | 0.009 | 0.013 | **0.017** | 0.961 |
| $\mathcal{M}$-flow (OT) | 0.089 | 0.433 | 0.134 | 0.647 |
| $\mathcal{M}$-flow (alternating OT/D) | 0.142 | 1.121 | 0.051 | 0.584 |
| $\mathcal{M}_e$-flow (S) | 0.005 | 0.006 | 0.033 | 0.975 |
| $\mathcal{M}_e$-flow (alternating M/D) | **0.003** | **0.003** | 0.030 | 0.985 |
| $\mathcal{M}_e$-flow (sequential M/D) | **0.002** | **0.002** | **0.007** | **0.987** |

**Table 2.** Results for the mixture model on a polynomial surface. We compare the sample quality of the different flows as given by their distance from the true data manifold (lower is better), the reconstruction error when projecting on the learned manifold (lower is better), the maximum mean discrepancy between MCMC samples based on the model and MCMC based on the true likelihood (lower is better), and the AUC when discriminating test samples from a second out-of-distribution test set (higher is better). Out of five runs with independent training data and initializations we show the median. The best four results, which are generally consistent with each other within the variance observed in the five runs, are shown in bold.

**Fig. 8.** Lorenz attractor. **Left**: Three example trajectories in the Lorenz system. They start very close to each other, soon diverge, but all tend to the Lorenz attractor. The black dots show points sampled from this system as described in the text. **Center and right**: manifold and probability density learned by an $\mathcal{M}$-flow model. Each dot represents a test sample projected onto the learned manifold, with the color indicating the learned log likelihood (bright yellow represents a high density, dark blue a low density). Both panels show the same model from different perspectives.

under the three ordinary differential equations

$$\frac{\mathrm{d}x_0}{\mathrm{d}t} = \sigma(x_1 - x_0)\,,$$

$$\frac{\mathrm{d}x_1}{\mathrm{d}t} = x_0(\rho - x_2) - x_1\,,$$

$$\frac{\mathrm{d}x_2}{\mathrm{d}t} = x_0 x_1 - \beta x_2\,. \tag{32}$$

For the canonical parameter choices $\sigma = 10$, $\beta = \frac{8}{3}$, and $\rho = 28$, the Lorenz system has chaotic solutions, and from many initial conditions the system will tend to the *Lorenz attractor*. This strange attractor has a Hausdorff dimension of approximately 2.06 (31) and admits an ergodic invariant probability measure (32).

We train an $\mathcal{M}$-flow model to learn the invariant probability density of the Lorenz attractor on a two-dimensional manifold. We generate 100 trajectories with different initial conditions, evolving each from $t = 0$ to $t = 1000$. Then we sample positions $x_i$ uniformly over all trajectories and time (except that we skip a warm-up period from $t = 0$ to $t = 50$). Example trajectories and samples are shown in the left panel of Figure 8. On these i.i.d. samples we train an $\mathcal{M}$-flow model based on a rational-quadratic neural spline flow architecture. The system and the flow architecture are described in detail in the appendix.

The learned manifold and density is shown in the middle and right panels of Figure 8. The $\mathcal{M}$-flow succesfully learned the overall shape of the attractor, including the nontrivial part with two disconnected manifold branches, as well as a plausible probability density on that manifold. The model is not perfect, with some artifacts likely linked to the nontrivial topology of the manifold. Nevertheless, the results demonstrate that $\mathcal{M}$-flows are a useful off-the-shelf method even for problems with a nontrivial topology.

**D. Particle physics.** Our next experiment is a real-world problem from particle physics, in which the goal is to infer fundamental constants of Nature from data collected in proton-proton collisions at the Large Hadron Collider experiments. We study a process in which a Higgs boson is produced in the "weak boson fusion" mode and decays into two photons. While

the raw data measured in such a process consists of hundreds of millions of sensor readouts, we follow the common strategy to reduce the data to an unstructured vector of 40 features. Our model of the process is based on a real-world simulator described in the appendix, which takes a three-dimensional parameter point $\theta \in \mathbb{R}^3$ as input and samples synthetic data $x \in \mathbb{R}^{40}$ according to an implicit density $p^*(x|\theta)$.

The likelihood function of the simulator is intractable, our goal is thus to infer the posterior over $\theta$ given an observed dataset based only on training samples and the corresponding parameter points. This setting is known as likelihood-free or simulation-based inference (33). However, the process is not entirely a black box and domain experts can contribute helpful insight about the structure of the data: from the laws of particle physics and the definition of the summary statistics we know that the data populate some 14-dimensional manifold within the 40-dimensional data space. This makes it an ideal test case for our $\mathcal{M}$-flow setup. The shape of the manifold is independent of the model parameters $\theta$, while the probability density on the manifold is conditional on them. While the effects of these parameters on the probability density consist of subtle variations that may not be easy to pick up for probabilistic models, resolving them is essential for scientific inference in particle physics experiments.

We follow a neural likelihood estimation strategy (34, 35) and train conditional density estimators on $10^6$ samples generated from the simulator to learn the likelihood function. We consider $\mathcal{M}$-flow, $\mathcal{M}_e$-flow, PIE, PAE, and AF models based on neural spline flows, using 35 coupling layers interspersed with linear transformations. More details are given in the appendix. Since the adversarial training did not lead to satisfactory results even in the low-dimensional experiments, we do not include it here.

In addition to the regular training of these algorithms based on maximum likelihood and the M/D schedules, we train models with the SCANDAL method introduced in Refrence (36). We extract additional information from the simulator that characterizes its latent process and augment the training data with it. We then add a term to the loss functions[§] that in-

---
[§]The SCANDAL loss term is only added to the likelihood-based loss functions, not to the manifold

Brehmer and Cranmer

**Fig. 9.** Inference results on the particle physics problem. We show posterior samples from an MCMC sampler based on likelihood estimates from different flow models. The rows are based on different synthetic observations, the stars mark the true parameter points $\theta^*$ used to generate the observation data. The first four columns use models trained by maximum likelihood and the M/Dalgorithm, the columns marked with "(S)" use the SCANDAL method of augmenting the training data. The rightmost column ("LR (A)") shows the results from a likelihood ratio estimator trained with the ALICES method.

| Model (algorithm) | Sample closure | Mean reconstruction error | Log posterior |
|---|---|---|---|
| AF | **0.0019** $\pm$ 0.0001 | – | $-3.94 \pm 0.87$ |
| PIE (original) | 0.0023 $\pm$ 0.0001 | 2.054 $\pm$ 0.076 | $-4.68 \pm 1.56$ |
| PIE (unconditional manifold) | 0.0022 $\pm$ 0.0001 | 1.681 $\pm$ 0.136 | $-1.82 \pm 0.18$ |
| PAE | 0.0073 $\pm$ 0.0001 | 0.052 $\pm$ 0.001 | – |
| $\mathcal{M}$-flow | 0.0045 $\pm$ 0.0004 | **0.012** $\pm$ 0.001 | $-1.71 \pm 0.30$ |
| $\mathcal{M}_e$-flow | 0.0046 $\pm$ 0.0002 | 0.029 $\pm$ 0.001 | $-1.44 \pm 0.34$ |
| AF (SCANDAL) | 0.0565 $\pm$ 0.0059 | – | $-0.40 \pm 0.09$ |
| PIE (original, SCANDAL) | 0.1293 $\pm$ 0.0218 | 3.090 $\pm$ 0.052 | 0.03 $\pm$ 0.17 |
| PIE (uncond. manifold, SCANDAL) | 0.1019 $\pm$ 0.0104 | 1.751 $\pm$ 0.064 | **0.23** $\pm$ 0.05 |
| PAE (SCANDAL) | 0.0323 $\pm$ 0.0010 | 0.053 $\pm$ 0.001 | – |
| $\mathcal{M}$-flow (SCANDAL) | 0.0371 $\pm$ 0.0030 | **0.011** $\pm$ 0.001 | 0.11 $\pm$ 0.04 |
| $\mathcal{M}_e$-flow (SCANDAL) | 0.0291 $\pm$ 0.0010 | 0.030 $\pm$ 0.002 | **0.14** $\pm$ 0.09 |
| Likelihood ratio estimator (ALICES) | – | – | 0.05 $\pm$ 0.05 |

**Table 3.** Results for the particle physics dataset. We compare the sample quality of the different flows as given by sample closure defined in Eq. (33) (lower is better). As second metric we give the mean reconstruction error when projecting test samples to the learned manifold (lower is better). Finally, the performance on an inference task is evaluated with the log posterior of the true parameter point (see text, higher is better). For each mode, we train at least five instances with independent initializations, remove the top and bottom value, and report the mean over the remaining runs. The best results are shown in bold.

centivizes the score $\nabla_\theta \log p(x|\theta)$ of the flow to be accurate, ultimately improving the sample efficiency and quality of inference.

Finally, as an additional inference baseline we consider a neural estimator of the likelihood *ratio* $p(x|\theta)/p(x|\theta_{\mathrm{ref}})$ between two different parameter points. It is based on the decision function of a classifier, an approach known as the *likelihood ratio trick* (37). Intuitively, a likelihood ratio estimator does not have to model the probability density everywhere in space and may thus have an easier time learning the subtle effects of the parameters $\theta$ on the statistical model. We train it with the ALICES technique, which similarly to the SCANDAL loss discussed above leverages additional information that characterizes the latent process in the simulator (38). Likelihood ratio estimators are known to provide a strong baseline for inference, but lose the ability to generate data (33, 39).

The models are first evaluated through their generative mode, gauging the quality of samples with a set of closure

<span style="font-size:small">updates in the M/D training.</span>

tests. From domain knowledge we can construct a series of constraints $c_i(x)$, which for samples from nature (or our simulator) almost vanish, $c_i(x) \approx 0$. Small deviations from zero can arise from machine precision issues and, in some cases, from experimental noise in the detector model. For each trained generative model $p(x|\theta)$ we calculate the mean sum of deviations from these conditions as the *sample closure*

$$\mathbb{E}_{x \sim p(x|\theta)}\left[ w_i \sum_i |c_i(x)| \right]. \qquad [33]$$

Here the weights $w_i$ of the individual closure tests are chosen such that for a product of the marginal densities (corresponding to a random reshuffling of all features across samples) the sample closure is equal to 1 and has equal contribution from each term in the sum. This quantity gives us a metric of sample quality, with smaller values being better and an expected range from 0 to 1. In addition, we evaluate the quality of the learned manifold of the PIE and $\mathcal{M}$-flow models by project-

**Fig. 10.** Two-dimensional StyleGAN image manifolds. **Left**: true image manifold parameterized by the latent variables of the StyleGAN (these latent variables are never available to the flow models during training). **Center**: image manifold learned by an $\mathcal{M}$-flow model. **Right**: image manifold learned by an $\mathcal{M}_e$-flow model.

ing test samples to the learned manifold and computing the corresponding reconstruction error.

As a final metric we study the quality of inference on the parameters $\theta$ given an observed set of 20 observed samples $\{x_{\text{obs}}\} \sim p(x|\theta^*)$. As in the previous experiment, we use the different flow models as a surrogate for the likelihood in a Metropolis-Hastings MCMC sampler to generate posterior samples $\{\theta\} \sim p(\theta|\{x_{\text{obs}}\})$. As the implicit density of the simulator is intractable, we cannot compare to the true posterior. Instead we use kernel density estimation to evaluate the posterior based on the different models at the ground-truth parameter point $\theta^*$. For each model, we compute the posterior of three different true parameter points $\theta^*$ and report the average of the log posterior over the three values.

In Table 3 and Figure 9 we show that the AF models produce the most realistic samples according to the closure test, but result in unreliable inference results. Pure likelihood-based training seems to inventivize learning the overall data distribution more than figuring out the subtle effects of the parameters of interest. Our improved PIE version with an unconditional manifold as well as the new $\mathcal{M}$-flow and $\mathcal{M}_e$-flow models perform better in the inference task. Moreover, the $\mathcal{M}$-flow and $\mathcal{M}_e$-flow models define higher-quality manifolds than the PIE and PAE models as measured by the reconstruction error when projecting test samples to the learned manifolds. Training any model on the SCANDAL loss reduces the sample quality according to the closure test, but substantially improves the quality of inference to be on par or even better than the likelihood ratio estimator baseline.

**E. StyleGAN image manifolds.** As a first application to images, we study synthetic datasets in which the images populate a manifold of known dimension $n$. To this end we generate data from a state-of-the-art GAN, a StyleGAN2 (40) model trained on the FFHQ dataset (41). By sampling $n$ of the latent variables of this model while keeping all other latent variables fixed, we generate training and evaluation data that populate an $n$-dimensional manifold. We construct two such datasets, using $n = 2$ for the first and $n = 64$ for the second. In the second dataset, the distribution of images on the manifold depends on a model parameter $\theta$, which scales the variance with which the GAN latent variables are sampled. In both cases we downsample the images to a resolution of $64 \times 64$. We illustrate the two-dimensional image manifold in the left panel of Figure 10.

We then train $\mathcal{M}$-flow, $\mathcal{M}_e$-flow, PIE, and AF models on a training set of 10 000 images for the two-dimensional manifold and 20 000 images for the 64-dimensional manifold. Again we implement the models as rational-quadratic neural spline flows. The transformations $f$ are based on a multi-scale architecture with 20 ($\mathcal{M}$-flow, $\mathcal{M}_e$-flow, PIE) or 28 (AF) coupling layers across four levels interspersed with actnorm layers and $1 \times 1$ convolutions (5, 28, 42). In the $\mathcal{M}$-flow, $\mathcal{M}_e$-flow, and PIE models we apply two additional invertible linear layers to a subset of the channels before projecting to the manifold coordinates $u$; this gives the models the freedom to align the learned manifold with features across different scales. The latent transformation $h$ is implemented as a sequence of 8 coupling layers alternating with invertible linear layers. Architectures and training are described in detail in the appendix.

| Model | $n = 2$ | | $n = 64$ | | |
| --- | --- | --- | --- | --- | --- |
| | FID score | Mean reconstruction error | FID score | Mean reconstruction error | Log posterior |
| AF | $58.3 \pm 1.5$ | – | $24.0 \pm 0.0$ | – | $0.17 \pm 1.18$ |
| PIE | $139.5 \pm 5.0$ | $5539 \pm 56$ | $32.2 \pm 0.8$ | $4155 \pm 31$ | $-6.40 \pm 1.54$ |
| $\mathcal{M}$-flow | $43.9 \pm 0.2$ | $332 \pm 9$ | $\mathbf{20.8} \pm 0.5$ | $\mathbf{1430} \pm 4$ | $\mathbf{2.67} \pm 0.27$ |
| $\mathcal{M}_e$-flow | $\mathbf{43.5} \pm 0.2$ | $\mathbf{303} \pm 4$ | $23.7 \pm 0.2$ | $1555 \pm 3$ | $1.81 \pm 0.70$ |

**Table 4.** Results for the StyleGAN image manifolds. We evaluate the quality of samples generated by the different flows through the Fréchet Inception Distance (lower is better). In addition, we report the mean reconstruction error when projecting test samples to the learned manifold (lower is better). For each algorithm, we train ten instances with independent initializations, remove the top and bottom value, and report the mean between the eight remaining runs. The best results are shown in bold.

**Fig. 11.** StyleGAN image manifold samples. The top rows shows test samples, below we show samples generated from various flow models. For PIE we show both samples drawn from the learned manifold with $v = 0$ ("manifold"), as well as samples drawn with $v \sim p_v(v)$, which extends off the manifold and corresponds to the PIE likelihood ("off-manifold"); see Section 2.C for a discussion. **Left**: $n = 2$, curated from a pool of 20 random samples, aiming for a diverse selection. **Right**: $n = 64$, uncurated samples.

Figure 11 shows samples generated from the various models. Subjectively, the $\mathcal{M}$-flow and $\mathcal{M}_e$-flow samples look most realistic and diverse; the AF model comes close. When restricted to sampling from the manifold, as originally proposed in Reference (21), the PIE models suffer from mode collapse. This is particularly pronounced in the $n = 2$ case, where the model samples the same face over and over again; clearly the learned manifold (the level set $v = 0$) is not aligned with the true image manifold at all. When we sample the off-the-manifold coordinates $v \sim p_v(v)$ as well, the quality and diversity increases, though the images are still of a poorer quality than those from the $\mathcal{M}$-flow, $\mathcal{M}_e$-flow, and AF models.

In the center and right panels of Figure 10 we show the image manifold learned by an $\mathcal{M}$-flow and an $\mathcal{M}_e$-flow model from the $n = 2$ dataset. Both learn a manifold that covers virtually identical images to the ground truth. In contrast, the PIE models (not shown) do not learn a useful manifold and instead assign most of the variance in the image distribution to the "off-the-manifold" coordinates $v$. Note that the $\mathcal{M}$-flow and $\mathcal{M}_e$-flow models all learn different charts for the same manifold: a given image corresponds to different latent variable vector $\tilde{u}$ in each model, and all of them differ from the latent variables of the StyleGAN that generated the training data. This is just as expected, since manifolds are invariant under

reparameterizations. Finally, all learned charts are smooth in the sense that images change gradually when moving along the manifold coordinates.

Next, we demonstrate the faithfulness of the learned image manifolds by projecting test images to the learned image manifolds and comparing the projected image to the original. As shown in Figure 12, the $\mathcal{M}$-flow and $\mathcal{M}_e$-flow projections are very faithful, with small differences visible in fine details like in the hair. As anticipated from the previous results, the PIE manifolds are not very helpful.

In Table 4, the models are evaluated based on the Fréchet Inception Distance (FID score) (43, 44) between samples generated from them and test samples drawn from the "true" GAN model, and on the reconstruction error when projecting test samples to the learned manifold. We train 10 ($n = 2$) or 3 ($n = 64$) instances of each architecture with independent random seeds and report the mean and its error of the remaining eight runs. On all metrics, the $\mathcal{M}$-flow and $\mathcal{M}_e$-flow models clearly outperform the AF and PIE baselines.

**F. Real-world images.** Finally, we train $\mathcal{M}$-flow, $\mathcal{M}_e$-flow, AF, and PIE models on the real-world image dataset CelebA-HQ (41), downsampled to a resolution of $64 \times 64$. Unlike in the previous datasets, the existence of a data manifold and

**Fig. 12.** Projections to StyleGAN image manifold. We show test images (top row), their projection to the learned manifold in three models, and the residuals (white corresponds to perfect reconstructions). **Left**: $n = 2$. **Right**: $n = 64$.

its dimension are not known. For this first exploration we use $\mathcal{M}$-flow and $\mathcal{M}_e$-flow models with manifold dimensionality $n = 512$, leaving a systematic determination of the properties of the image manifold (if it exists) for future work. In all other respects, we use the same architecture as in the previous section.

We show samples from the different models in Figure 13 and compare FID scores and reconstruction errors from the projection to the manifold in Table 5. While the $\mathcal{M}$-flow and $\mathcal{M}_e$-flow models learn a higher-quality manifold than PIE, the AF models produce the most realistic images. This may point to a suboptimal choice of the manifold dimension $n$ in this proof-of-principle study.

In Figure 14 we sketch how the performance of the $\mathcal{M}$-flow model on CelebA varies with the choice of the manifold dimension $n$. We only show one run per choice of $n$ and only train these models for a limited number of steps, so these results are mostly illustrative. Nevertheless, our result provide a hint for $\mathcal{M}$-flow models to stabilize with $n \gtrsim 100$.

## 5. Related work

Our work is closely related to a number of different probabilistic and generative models. We have discussed the relation to normalizing flows, autoencoders, variational autoencoders, generative adversarial networks, and energy-based models in the introduction and in Section 2. In addition, manifold learning is its own research field with a rich set of methods (45), though these typically do not model the data density on the manifold and thus do not serve quite the same purpose as the models discussed in this paper. In the following we want to draw attention to three particularly closely related works and describe how our approach differs from them.

**Relaxed injective probability flows.** The work most closely related to manifold-learning flows are relaxed injective probability flows (10), which appeared while this paper was in its final stages of preparation. The proposed model is similar to our manifold-learning flows with a separate encoder ($\mathcal{M}_e$-flow). A key difference is the way in which the invertibility of the decoder $g$ is enforced. The authors of Reference (10) bound

**Fig. 13.** CelebA test samples (top row) and samples generated from different flow models. The shown samples were selected from a batch of 40 uncurated images.

the norm of the Jacobian of an otherwise unrestricted transformation $g$. While this makes the transformation in principle invertible (up to the possibility of multiple points in latent space pointing to the same point in data space), the inverse of $g$ and the likelihood of this model are not tractable for unseen data points. This makes their algorithm unsuitable for inference tasks. As the authors point out, their model also cannot deal with points off the learned manifold. We address these issues by drawing from the flow literature and defining the decoder as the level set of a diffeomorphism, which is by construction exactly invertible. We also add a prescription for evaluating off-the-manifold points with a projection to the manifold, which naturally provides a measure of distance between the data point and the manifold.

Similar to our discussion in Section A, the authors of Reference (10) also argue that training an injective flow by maximum likelihood is infeasible due to the computational cost of evaluating the Jacobian of $g$. They propose a different training objective that is based on a stochastic approximation of a lower bound of the likelihood, which can be computed efficiently. We point to this training strategy in our discussion in Section B, but realized that the alternating procedure allows us to sidestep the problem. Finally, their motivation is different from ours: while we develop $\mathcal{M}$-flows specifically to better represent the true structure of the data, they focus on the reduced computational complexity of the model due to a lower-dimensional space; they view the lack of support of the model off the manifold as a deficiency rather than an advantage. In addition to these qualitative differences, it would be interesting to compare relaxed injective probability flows and manifold-learning flows quantitatively.

**Pseudo-invertible encoder.** Another closely related model is the pseudo-invertible encoder (PIE) (21), which we define and discuss in Section 2 and use as a baseline in our experiments. The key difference to our $\mathcal{M}$-flow setup is that the PIE model describes a density over the ambient data space, while $\mathcal{M}$-flow limits the density strictly to the manifold. In this sense the PIE approach is much more similar to a standard ambient flow, though it adds a multi-scale architecture and different base densities for the latent variables that correspond to the manifold coordinates and the off-the-manifold latents. In addition to this fundamental difference in construction, PIE and $\mathcal{M}$-flow models are trained differently: for PIE maximum likelihood is sufficient, while for $\mathcal{M}$-flow we discuss the shortcomings of that objective and propose several new training schemes.

**Flows on manifolds.** Finally, the $\mathcal{M}$-flow is closely related to normalizing flows on (prescribed) manifolds (8) (FOM). In particular, the likelihood equation is almost the same, with the crucial exception that manifold flows require knowing a parameterization of the manifold in terms of coordinate and a chart, while the $\mathcal{M}$-flow algorithm learnes these from data. Since in many real-world cases the manifold is not known, $\mathcal{M}$-flow models are applicable to a much larger class of problems than FOM.

**Our contributions.** This paper contains four main contributions:

1. We propose two types of manifold-learning flows, $\mathcal{M}$-flows and $\mathcal{M}_e$-flows.

2. We identify a subtlety in the naive interpretation of the density of such models and argue that they should not be trained by naive maximum likelihood alone. We address this issue with the new manifold / density (M/D) training strategy, which separates manifold and density updates. This both reduces the computational cost of the likelihood evaluation during training as well as avoids issues with potential pathological configurations. We also discuss training strategies based on adversarial training and optimal transport.

3. We demonstrate these models and training algorithms in a range of experiments.

4. Beyond the newly proposed $\mathcal{M}$-flow models we provide a general discussion of the relation between different generative models and the data manifold, reviewing ambient flows, injective flows, flows on manifolds, PIEs, VAEs, and GANs in a common language. In particular, we identify an inconsistency between training and data generation for PIE models, and introduce a conditional PIE version

| Model | FID score | Reconstruction error |
|---|---|---|
| AF | **33.6** $\pm$ 0.2 | – |
| PIE | 75.7 $\pm$ 5.1 | 6970 $\pm$ 97 |
| $\mathcal{M}$-flow | 37.4 $\pm$ 0.2 | **830** $\pm$ 5 |
| $\mathcal{M}_e$-flow | 35.8 $\pm$ 0.4 | 991 $\pm$ 4 |

**Table 5.** Results for the CelebA dataset. We evaluate the quality of samples generated by the different flows through the Fréchet Inception Distance (lower is better). For each algorithm, we train three instances with independent initializations and report the mean. The best results are shown in bold.

**Fig. 14.** CelebA performance of the $\mathcal{M}$-flow model as a function of the manifold dimension $n$. We show FID scores (dashed purple, left axis) and the mean reconstruction error from projecting to the manifold (solid orange, right axis).

for the case where only the density, but not the manifold shape, depends on the parameters being conditioned on.

## 6. Conclusions

In this work we introduced manifold-learning flows ($\mathcal{M}$-flows), a new type of generative model that combines aspects of normalizing flows, autoencoders, and GANs. $\mathcal{M}$-flows describe data as a probability density over a lower-dimensional manifold embedded in data space. Unlike flows on prescribed manifolds, they learn a chart for the manifold from the training data. $\mathcal{M}$-flows allow generating data in a similar way to GANs while maintaining a tractable exact density. They also provide a prescription for evaluating points off the manifold by first projecting data onto the manifold. The $\mathcal{M}$-flow approach may not only represent datasets with manifold structure more accurately, but also allow us to use lower-dimensional latent spaces than with ambient flows, reducing the memory and computational footprint. As an added benefit, the projection to the manifold may be useful for denoising or to detect out-of-distribution samples. We introduced two variants of this new model, one of which features a separate encoder while the other uses the inverse of the decoder directly, and broadly reviewed the relation between several types of generative models and the structure of the data manifold.

Despite the tractable density, training $\mathcal{M}$-flow models is nontrivial: any update of the manifold modifies the data variable that the density is describing, rendering training by naive maximum likelihood invalid. In addition, computing the full model likelihood can be expensive. We reviewed several training and evaluation strategies that mitigate this problem. In particular, we introduced the new M/D training schedule, which separates manifold and density updates and solves both stability and training issues. We also presented an adversarial training scheme based on optimal transport as well as a hybrid version that alternates between adversarial phases and density updates.

In a first suite of experiments ranging from simple pedagogical examples to a real-world physics example to image datasets, we demonstrated how this approach lets us learn the data manifold and a probability density on it. $\mathcal{M}$-flows learned manifolds of a higher quality than PIE baselines, and performed better than ambient flow and PIE models on most downstream inference tasks and many of the generative metrics we considered.

Problems in which data populates a lower-dimensional manifold embedded in a high-dimensional feature space are almost everywhere. In some scientific cases, domain knowledge allows for exact statements about the dimensionality of the data manifold, and $\mathcal{M}$-flows can be a particularly powerful tool in a likelihood-free or simulation-based inference setting (33). Even in the absence of such domain-specific insight this approach may be valuable: GANs with low-dimensional latent spaces are powerful generative models for numerous datasets of natural images, which is testament to the presence of a low-dimensional data manifold. Flows that simultaneously learn the data manifold and a tractable density over it may help us to unify generative and inference tasks in a way that is tailored to the structure of the data.

**Acknowledgements.** We would like to thank Jens Behrmann, Kyunghyun Cho, Jack Collins, Jean Feydy, Siavash Golkar, Michael Kagan, Dimitris Kalatzis, Gilles Louppe, George Papamakarios, Merle Reinhart, Frank Rösler, John Tamanas, Antoine Wehenkel, and Andrew Wilson for useful discussions. We are grateful to Conor Durkan, Artur Bekasov, Iain Murray, and George Papamakarios for publishing their excellent neural spline flow codebase (28), which we used extensively in our analysis. Similarly, we want to thank George Papamakarios, David Sterratt, and Iain Murray for publishing their Sequential Neural Likelihood code (34), parts of which were used in the evaluation steps in our experiments.

We are grateful to the authors and maintainers of DELPHES 3 (46), GEOMLOSS (24), JUPYTER (47), MAD-GRAPH5_AMC (48), MADMINER (49), MATPLOTLIB (50), NUMPY (51), PYTHIA8 (52), PYTORCH (53), PYTORCH-FID (54), SCIKIT-LEARN (55), and SCIPY (56). We are grateful for the support of the National Science Foundation under the awards ACI-1450310, OAC-1836650, and OAC-1841471, as well as by the Moore-Sloan data science environment at NYU. This work was supported in part through the NYU IT High Performance Computing resources, services, and staff expertise; through the NYU Courant Institute of Mathematical Sciences; and by the Scientific Data and Computing Center at Brookhaven National Laboratory.

1. Goodfellow IJ, et al. (2014) Generative adversarial nets in *Advances in Neural Information Processing Systems*.
2. Kingma DP, Welling M (2014) Auto-encoding variational bayes in *2nd International Conference on Learning Representations, ICLR 2014 - Conference Track Proceedings*.
3. Dinh L, Krueger D, Bengio Y (2015) NICE: Non-linear independent components estimation. *3rd International Conference on Learning Representations, ICLR 2015 - Workshop Track Proceedings*.
4. Rezende DJ, Mohamed S (2015) Variational inference with normalizing flows in *32nd International Conference on Machine Learning, ICML 2015*.
5. Dinh L, Sohl-Dickstein J, Bengio S (2019) Density estimation using real NVP in *5th International Conference on Learning Representations, ICLR 2017 - Conference Track Proceedings*.
6. Papamakarios G, Nalisnick E, Jimenez Rezende D, Mohamed S, Lakshminarayanan B (2019) Normalizing Flows for Probabilistic Modeling and Inference. *arXiv:1912.02762*.
7. Feinman R, Parthasarathy N (2019) A Linear Systems Theory of Normalizing Flows. *arXiv:1907.06496*.
8. Gemici MC, Rezende D, Mohamed S (2016) Normalizing Flows on Riemannian Manifolds. *arXiv:1611.02304*.

9. Teng Y, Choromanska A, Bojarski M (2018) Invertible Autoencoder for domain adaptation. *arXiv:1802.06869*.
10. Kumar A, Poole B, Murphy K (2020) Regularized Autoencoders via Relaxed Injective Probability Flow. *arXiv:2002.08927*.
11. Ghosh P, Sajjadi MS, Vergari A, Black M, Schölkopf B (2019) From Variational to Deterministic Autoencoders. *arXiv:1903.12436*.
12. LeCun Y, Chopra S, Hadsell R, Ranzato M, Huang F (2006) A tutorial on energy-based learning. *Predicting structured data* 1(0).
13. Che T, et al. (2020) Your GAN is Secretly an Energy-based Model and You Should use Discriminator Driven Latent Sampling. *arXiv:2003.06060*.
14. Arbel M, Zhou L, Gretton A (2020) KALE: When Energy-Based Learning Meets Adversarial Training. *arXiv:2003.05033*.
15. Arjovsky M, Chintala S, Bottou L (2017) Wasserstein GAN. *arXiv:1701.07875*.
16. Rezende DJ, et al. (2020) Normalizing Flows on Tori and Spheres. *arXiv:2002.02428*.
17. Bose AJ, Smofsky A, Liao R, Panangaden P, Hamilton WL (2020) Latent Variable Modelling with Hyperbolic Normalizing Flows. *arXiv:2002.06336*.
18. Kanwar G, et al. (2020) Equivariant flow-based sampling for lattice gauge theory. *arXiv:2003.06413*.
19. Ramesh A, LeCun Y (2018) Backpropagation for Implicit Spectral Densities. *arXiv:1806.00499*.
20. Dieng AB, Ruiz FJR, Blei DM, Titsias MK (2019) Prescribed Generative Adversarial Networks. *arXiv:1910.04302*.
21. Beitler JJ, Sosnovik I, Smeulders A (2019) {PIE}: Pseudo-Invertible Encoder.
22. Böhm V, Seljak U (2020) Probabilistic auto-encoder. *arXiv:2006.05479*.
23. Grover A, Dhar M, Ermon S (2017) Flow-GAN: Combining Maximum Likelihood and Adversarial Learning in Generative Models. *arXiv:1705.08868*.
24. Feydy J, et al. (2019) Interpolating between Optimal Transport and MMD using Sinkhorn Divergences in *The 22nd International Conference on Artificial Intelligence and Statistics*. pp. 2681–2690.
25. Gropp A, Yariv L, Haim N, Atzmon M, Lipman Y (2020) Implicit Geometric Regularization for Learning Shapes. *arXiv:2002.10099*.
26. Behrmann J, Grathwohl W, Chen RT, Duvenaud D, Jacobsen JH (2018) Invertible Residual Networks. *arXiv:1811.00995*.
27. Chen RT, Behrmann J, Duvenaud D, Jacobsen JH (2019) Residual Flows for Invertible Generative Modeling. *arXiv:1906.02735*.
28. Durkan C, Bekasov A, Murray I, Papamakarios G (2019) Neural Spline Flows. *Advances in Neural Information Processing Systems* pp. 7509–7520.
29. Gretton A, Borgwardt KM, Rasch MJ, Schölkopf B, Smola A (2012) A kernel two-sample test. *Journal of Machine Learning Research* 13(Mar):723–773.
30. Lorenz EN (1963) Deterministic Nonperiodic Flow. *Journal of the Atmospheric Sciences* 20(2):130–141.
31. Viswanath D (2004) The fractal property of the Lorenz attractor. *Physica D: Nonlinear Phenomena* 190:115–128.
32. Guckenheimer J, Sparrow C (1984) The Lorenz Equations: Bifurcations, Chaos, and Strange Attractors. *The American Mathematical Monthly*.
33. Cranmer K, Brehmer J, Louppe G (2020) The frontier of simulation-based inference in *Proceedings of the National Academy of Sciences*. (National Academy of Sciences).
34. Papamakarios G, Sterratt DC, Murray I (2018) Sequential Neural Likelihood: Fast Likelihood-free Inference with Autoregressive Flows.
35. Lueckmann JM, Bassetto G, Karaletsos T, Macke JH (2018) Likelihood-free inference with emulator networks.
36. Brehmer J, Louppe G, Pavez J, Cranmer K (2020) Mining gold from implicit models to improve likelihood-free inference. *Proceedings of the National Academy of Sciences of the United States of America* 117(10):5242–5249.
37. Cranmer K, Pavez J, Louppe G (2015) Approximating Likelihood Ratios with Calibrated Discriminative Classifiers. *arXiv:1506.02169*.
38. Stoye M, Brehmer J, Louppe G, Pavez J, Cranmer K (2019) Likelihood-free inference with an improved cross-entropy estimator. *NeurIPS workshop on Machine Learning for the Physical Sciences*.
39. Durkan C, Murray I, Papamakarios G (2020) On Contrastive Learning for Likelihood-free Inference. *arXiv:2002.03712*.
40. Karras T, et al. (2019) Analyzing and Improving the Image Quality of StyleGAN. *arXiv:1912.04958*.
41. Karras T, Laine S, Aila T (2018) A Style-Based Generator Architecture for Generative Adversarial Networks. *arXiv:1812.04948*.
42. Kingma DP, Dhariwal P (2018) Glow: Generative flow with invertible 1×1 convolutions. *Advances in Neural Information Processing Systems* 2018-Decem:10215–10224.
43. Heusel M, Ramsauer H, Unterthiner T, Nessler B, Hochreiter S (2017) GANs Trained by a Two Time-Scale Update Rule Converge to a Local Nash Equilibrium. *arXiv:1706.08500*.
44. Lucic M, Kurach K, Michalski M, Gelly S, Bousquet O (2017) Are GANs Created Equal? A Large-Scale Study. *arXiv:1711.10337*.
45. Cayton L (2005) Algorithms for manifold learning. *Univ. of California at San Diego Tech. Rep* 12(1-17):1.
46. Demin P, Selvaggi M (2014) DELPHES 3, A modular framework for fast simulation of a generic collider experiment. *JHEP* 02:57.
47. Kluyver T, et al. (2016) Jupyter Notebooks - a publishing format for reproducible computational workflows in *ELPUB*.
48. Alwall J, et al. (2014) The automated computation of tree-level and next-to-leading order differential cross sections, and their matching to parton shower simulations. *Journal of High Energy Physics* 2014(7):79.
49. Brehmer J, Kling F, Espejo I, Cranmer K (2020) MadMiner: Machine Learning-Based Inference for Particle Physics. *Computing and Software for Big Science* 4(1).
50. Hunter JD (2007) Matplotlib: A 2D graphics environment. *Computing In Science & Engineering* 9(3):90–95.
51. van der Walt S, Colbert SC, Varoquaux G (2011) The NumPy Array: A Structure for Efficient Numerical Computation. *Computing in Science and Engineering* 13(2):22–30.
52. Sjöstrand T, Mrenna S, Skands P (2008) A brief introduction to PYTHIA 8.1. *Computer Physics Communications* 178(11):852–867.
53. Paszke A, et al. (2017) Automatic differentiation in PyTorch in *NIPS-W*.
54. Seitzer M (2020) Fréchet Inception Distance (FID score) in PyTorch.
55. Pedregosa F, et al. (2011) Scikit-learn: Machine Learning in {P}ython. *Journal of Machine Learning Research* 12:2825–2830.
56. Jones E, Oliphant T, Peterson P, Others (2001) {SciPy}: Open source scientific tools for {Python}.
57. Degrande C, et al. (2013) Effective field theory: A modern approach to anomalous couplings. *Annals of Physics* 335:21–32.

## A. Broader impact

Manifold-learning flows have the potential to improve the efficiency with which scientists extract knowledge from large-scale experiments. Many phenomena have their most accurate description in terms of complex computer simulations which do not admit a tractable likelihood. In this common case, normalizing flows can be trained on synthetic data and used as a surrogate for the likelihood function, enabling high-quality inference on model parameters (33). When the data have a manifold structure, manifold-learning flows may improve the quality and efficiency of this process further and ultimately contribute to scientific progress. We have demonstrated this with a real-world particle physics dataset, though the same technique is applicable to fields as diverse as neuroscience, systems biology, and epidemiology.

All generative models carry a risk of being abused for the generation of fake data that are then masqueraded as real documents. This danger also applies to manifold-learning flows. While manifold-learning flows are currently far away from being able to generate realistic high-resolution images, videos, or audio, this concern should be kept in mind in the long term.

Finally, the models we trained on image datasets of human faces clearly lack diversity. They reproduce and reinforce the biases inherent in the training data. Before using such (or other) models in any real-life application, it is crucial to understand, measure, and mitigate such biases.

## B. Experiment details

### A. Mixture model on a polynomial surface.

**Dataset.** In our second experiment, we consider a two-dimensional manifold embedded in three-dimensional Euclidean space defined by Eq. (30). We use the randomly drawn polynomial coefficients

$$
\begin{aligned}
f(z) = \exp(-0.1\|z\|) \big( &-1.217 + 1.522z_0 - 1.214z_1 \\
&+ 0.057z_0^2 - 0.024z_0z_1 - 0.047z_1^2 - 0.056z_0^3 - 0.008z_0^2 z_1 \\
&- 0.057z_0z_1^2 - 0.052z_1^3 + 0.014z_0^4 + 0.000z_0^3 z_1 - 0.007z_0^2 z_1^2 \\
&- 0.007z_0z_1^3 + 0.003z_1^4 - 0.008z_0^5 - 0.011z_0^4 z_1 + 0.004z_0^3 z_1^2 \\
&- 0.005z_0^2 z_1^3 - 0.009z_0z_1^4 + 0.012z_1^5 \big)
\end{aligned}
\quad [34]
$$

and the rotation matrix

$$
R = \begin{pmatrix} 0.974 & -0.227 & -0.009 \\ 0.227 & 0.973 & 0.040 \\ 0.000 & -0.041 & 0.999 \end{pmatrix} . \quad [35]
$$

For the training dataset we draw parameter points from a uniform prior, $\theta \sim \mathrm{Uniform}(-1, 1)$, while for the test set we generate data for $\theta = 0$.

**Architectures.** We study AF, PIE, $\mathcal{M}$-flow, and $\mathcal{M}_e$-flow models based on rational-quadratic neural spline flows, alternating coupling layers and random feature permutations [28]. For AF models we use ten coupling layers. For PIE, $\mathcal{M}$-flow, and $\mathcal{M}_e$-flow models we use five layers for the transformation $f$ (which also defines $g$ through a level set), and five layers for the transformation $h$. For the PIE model we use an off-the-manifold base density $p_v(v)$ with standard deviation $\epsilon = 0.01$. In each coupling transform, half of the inputs are elementwise transformed with a monotonic rational-quadratic spline, the parameters of which are determined from a residual network with two residual block of two hidden layers each, 100 units in each layer, and ReLU activations throughout. We do not use batch normalization or dropout since we found that the stochasticity they induce can lead to issues with the invertibility of the transformations. The splines are constructed in ten bins distributed over the range $(-6, 6)$.

**Training.** All models are trained with the ADAM optimizer, with an initial learning rate of $3 \cdot 10^{-4}$ and cosine annealing, and weight decay of $10^{-6}$. To balance the sizes of the various terms in the loss functions, we multiply them with different weights. For the manifold phase of the M/D training, we weight the mean reconstruction error with a factor 1000. In the S training, we use the the mean negative log likelihood (in nats) weighted with a factor of 0.1 plus the mean reconstruction error weighted with a factor of 1000. For OT training we multiply the Sinkhorn divergence (defined with $\varepsilon = 0.05$) with 10.

All models are trained for 50 epochs with a batch size of 100 (1000 for the OT training), corresponding to $5 \cdot 10^4$ updates ($5 \cdot 10^3$ for the OT training). For the M/D and OT/D training, we split the number of epochs evenly between the two phases. We study a sequential as well as an alternating version of the M/D algorithm, where in the latter case we alternate between training phases after every epoch. In all cases, we checkpoint the model weights after each epoch and revert to the version that leads to the smallest validation loss.

**Metrics.** We evaluate generated samples $x$ by undoing the rotation, $z' = R^{-1}x$, and evaluating the distance in $z'_2$ direction to the manifold as $|f((z'_0, z'_1)^T) - z'_2|$.

For the inference task we use a Metropolis-Hastings MCMC sampler based on the different flow likelihoods. We consider a synthetic "observed" dataset of 10 i.i.d. samples generated for $\theta^* = 0$. For each model, we generate an MCMC chain of length 5000, with a Gaussian proposal distribution with mean step size 0.15 and a burn in of 100 steps.

## B. Lorenz attractor.

**Dataset.** We study the invariant probability density of the Lorenz system defined in Equation Eq. [32]. To generate the training data, 100 trajectories are seeded with

$$x(0) \sim \mathcal{N}\left(x \middle| \begin{pmatrix} 1 \\ 1 \\ 1 \end{pmatrix}, \mathrm{diag}(0.1^2, 0.1^2, 0.1^2)\right) \qquad [36]$$

and forward-simulated in $0 \le t \le 1000$ using the Runge-Kutta method of order 5(4). Then $10^6$ samples $\{x\}$ are sampled: uniformly over the 100 trajectories, and uniformly over the time interval $50 \le t \le 1000$. The sampling ensures an i.i.d. dataset. Finally, the spatial positions $x$ are rescaled to zero

mean and unit variance. The beginning of a few trajectories and some samples after this procedure are shown in the left panel of Figure 8.

**Architectures.** We train an $\mathcal{M}$-flow model to learn the (two-dimensional) manifold and a probability density on it. Again the transformations $f$ and $h$ are implemented as rational-quadratic neural spline flows, each with five coupling layers interspersed with random feature permutations [28]. In each coupling transform, half of the inputs are elementwise transformed with a monotonic rational-quadratic spline, the parameters of which are determined from a residual network with two residual block of two hidden layers each, 100 units in each layer, and ReLU activations throughout. We do not use batch normalization or dropout. The splines are constructed in five bins distributed over the range $(-3, 3)$.

**Training.** We train the $\mathcal{M}$-flow model with the sequential M/D algorithm, using the ADAMW optimizer with an initial learning rate of $3 \cdot 10^{-4}$, cosine annealing, and weight decay of $10^{-4}$. During the manifold phase, the squared reconstruction error is multiplied by a factor of 1000, while for the density phase we just use the mean negative log likelihood (in bits per dimensions). The model is trained for 100 epochs (or $10^6$ gradient steps), split evenly between the manifold and density phase, with a batch size of 100. Again, we checkpoint the model weights after each epoch and revert to the version that leads to the smallest validation loss.

**Metrics.** Our analysis of the Lorenz attractor is purely qualitative. The learned manifold and probability density is shown in the center and right panels of Figure 8. Each dot corresponds to a test sample point projected to the learned manifold, with the color indicating the learned log likelihood.

## C. Particle physics.

**Dataset.** The particle physics experiment is based on Higgs production in the "weak boson fusion" mode with a decay into two photons. We generate synthetic data for this process with the simulators MADGRAPH5_AMC [48], PYTHIA8 [52], and DELPHES 3 [46]. Each generated sample is characterized by a vector of summary statistics $x \in \mathbb{R}^{40}$, for which we use the energy, momentum, pseudorapidity, invariant mass of the final-state particles, the reconstructed Higgs boson, and the dijet system, as well as the pseudorapidity gap between the two jets.

We consider this process in dependence of two model parameters $\theta$, the coefficients of an effective field theory with the dimension-six operators $\mathcal{O}_W$ and $\mathcal{O}_{\tilde{W}}$ in the basis of Reference [57]. For the training dataset we draw parameter points from a unit Gaussian prior.

**Architectures.** Our AF, PIE, $\mathcal{M}$-flow, and $\mathcal{M}_e$-flow models are again based on rational-quadratic neural spline flows with coupling layers alternating with invertible linear (LU-decomposed) transformations, largely following the setup described in Ref. [28]. For AF we use 35 coupling layers. For PIE, $\mathcal{M}$-flow, and $\mathcal{M}_e$-flow models we use 20 layers for $f$ (and thus also $g$) and 15 layers for $h$. In each coupling transformation, half of the inputs are elementwise transformed with a monotonic rational-quadratic spline, the parameters of which are determined from a residual network with two residual block of two hidden layers each, 100 units in each layer, and

ReLU activations throughout. Again we do not use batch normalization or dropout. The splines are constructed in 11 bins distributed over the range $(-10, 10)$. For the PIE model we use an off-the-manifold base density $p_v(v)$ with standard deviation $\epsilon = 0.1$.

The ALICES baseline consists of a simple multi-layer perceptron with 3 hidden layers of 100 units each and ReLU activations.

**Training.** All models are trained with the ADAMW optimizer, with an initial learning rate of $3 \cdot 10^{-4}$ and cosine annealing, and weight decay of $10^{-5}$. The $\mathcal{M}$-flow and $\mathcal{M}_e$-flow models are trained with the sequential M/D algorithm. To balance the sizes of the various terms in the loss functions, we multiply them with different weights. For likelihood-based training phases, we simply use the negative mean log likelihood (in bits per dimension). For the manifold phase of the M/D training we weight the mean reconstruction error with a factor 1000. In all SCANDAL versions, we add the mean squared error between the model score and the joint score (36) weighted by a factor of 2 to the loss. We train for 50 epochs with a batch size of 100 (corresponding to $5 \cdot 10^5$ gradient steps). Again, we checkpoint the model weights after each epoch and revert to the version that leads to the smallest validation loss.

**Metrics.** For the inference task we use a Metropolis-Hastings MCMC sampler based on the different flow likelihoods. We consider three synthetic "observed" datasets, each of which contains 15 i.i.d. samples. They are generated with the simulator for $\theta^* = (0, 0)$, which corresponds to the *Standard Model*, an established baseline parameter point; for $\theta^* = (0.5, 0)$; and for $\theta^* = (-1, -1)$. For each model and each observed dataset, we generate four MCMC chains of length 750 each, with a Gaussian proposal distribution with mean step size 0.15 and a burn in of 100 steps. We then use kernel density estimation to evaluate the posterior at the ground-truth parameter point $\theta^*$

$$\hat{p}(\theta^*|\{x_{\text{obs}}\}) = \mathbb{E}_{\theta \sim \text{MCMC}(\cdot|\{x_{\text{obs}}\})} \left[ K_\varepsilon(\theta - \theta^*) \right] , \qquad [37]$$

where $K_\varepsilon$ is a Gaussian kernel with bandwith $\varepsilon = 0.1$.

### D. StyleGAN image manifolds.

**Dataset.** Training and evaluation data are generated from a StyleGAN2 model (40). We use configuration f described in Reference (40) trained on the the FFHQ dataset (41). This GAN model uses 512 latent variables $z$ and a number of additional latent noise variables. We fix all latent variables except $z_{i<n}$ to a single random sampling from a normal distribution (for the regular latent variables we use a standard deviation of 0.05, for the noise variables of 1). Our dataset is then defined through the $n$ remaining latent variables $(z_0, z_1)$. We consider two datasets:

1. $n = 2$, sampling $z_0$ and $z_1$ from a unit Gaussian. We generate a training set of $10^4$ images.

2. $n = 64$, sampling the $z_{0...63}$ from a Gaussian with mean 0 and variance $\exp(\theta)^2$. The model parameter $\theta$ is drawn from a unit Gaussian. We generate a training set of $2 \cdot 10^4$ images.

All images are downsampled to a resolution of $64 \times 64$. The images thus populate a 2-dimensional or 64-dimensional manifold embedded in a $64 \times 64 \times 3$-dimensional ambient space. We preprocess the 8-bit training data through uniform dequantization (42).

**Architectures.** We consider AF, PIE, $\mathcal{M}$-flow, and $\mathcal{M}_e$-flow models based on rational-quadratic neural spline flows (28).

- The AF models closely follow the setup described in Reference (28), which in turn is based on the Glow (42) and RealNVP (5) architectures. A multi-scale setup (5) with four levels is used. On each level, seven steps are stacked. Each step entails an actnorm layer, an invertible $1 \times 1$ convolution, and a rational-quadratic coupling transformation. Overall there are thus 28 coupling transformation layers.

- $\mathcal{M}$-flow use a similar setup for the transformation $f$, except that each level only uses five steps. The output of this multi-scale transformation is then transformed with an invertible linear ($LU$-decomposed) layer, an invertible activation function, and another invertible linear layer, all of which only act on the first few channels per scale. This gives the network some flexibility to align the manifold with features across different scales, while keeping the model size managable. The output is flattened and projected to the $n$-dimensional latent space $U$. The transformation $h$ consists of 8 additional rational-quadratic coupling transformations with invertible linear ($LU$-decomposed) transformations. Overall this architecture thus involves 28 layers of coupling transformations.

- The $\mathcal{M}_e$-flow setup for $f$ and $h$ is identical to the $\mathcal{M}$-flow case. The encoder $e$ is implemented as a residual convolutional network with six residual blocks across three spatial resolutions, followed by a linear transformation to the $n$-dimensional latent space.

- Finally, the PIE model uses the same architecture as the $\mathcal{M}$-flow. The base density $p_v(v)$ has a standard deviation $\epsilon = 0.1$.

We never use batch normalization or dropout. The splines in the rational-quadratic coupling transformations are constructed in 11 bins distributed over the range $(-10, 10)$.

**Training.** All models are trained with the ADAMW optimizer, with an initial learning rate of $3 \cdot 10^{-4}$ and cosine annealing, and weight decay of $10^{-5}$. The $\mathcal{M}$-flow and $\mathcal{M}_e$-flow models are trained with the sequential M/D algorithm. Our loss is the negative mean log likelihood (in bits per dimension), except in the manifold phase of the M/D training, where we use the mean reconstruction error. For $n = 2$ we train for 100 epochs with a batch size of 25 (corresponding to $4 \cdot 10^4$ gradient steps); for $n = 64$ we use 200 epochs (or $1.6 \cdot 10^5$ gradient steps). The model weights are checkpointed after each epoch, in the end we revert to the version that leads to the smallest validation loss.

**Metrics.** Samples generated from the models are evaluated based on the Fréchet Inception Distance (FID score) (43, 44), using the (unofficial, but validated) PyTorch implementation of Reference (54). On the 64-dimensional dataset, we also consider inference on $\theta$ based an observed sample of 50 i.i.d. samples $\{x\} \sim p(x|\theta^*)$ with $\theta^* = 0$. For each model and each observed dataset, we generate an MCMC chain of length 400 each, with a Gaussian proposal distribution with mean step size 0.15 and a burn in of 50 steps. As in the particle physics experiment, we use kernel density estimation to evaluate the posterior at the ground-truth parameter point $\theta^*$.

## E. Real-world images.

**Dataset.** We use the CelebA-HQ (41) downsampled to a resolution of $64 \times 64$ as prepared by Reference (28), which contains 27000 training images and 3000 test images.

**Architectures.** We use the same setup as for the StyleGAN image manifolds, except that we consider $\mathcal{M}$-flows and $\mathcal{M}_e$-flows with manifold dimension $n = 512$.

**Training.** All models are trained with the ADAMW optimizer, with an initial learning rate of $3 \cdot 10^{-4}$ and cosine annealing, and weight decay of $10^{-5}$. The $\mathcal{M}$-flow and $\mathcal{M}_e$-flow models are trained with the sequential M/D algorithm. Our loss is the negative mean log likelihood (in bits per dimension), except in the manifold phase of the M/D training, where we use the mean reconstruction error. We train for 500 epochs with a batch size of 25 (corresponding to $5.4 \cdot 10^5$ gradient steps). The model weights are checkpointed after each epoch, in the end we revert to the version that leads to the smallest validation loss.

**Metrics.** Samples generated from the models are evaluated based on the Fréchet Inception Distance (FID score) (43, 44), using the (unofficial, but validated) PyTorch implementation of Reference (54).

## Footnotes

[1]Corresponding author. E-mail: johann.brehmer@nyu.edu.