[Reviews · NeurIPS 2020]

Review 1

Summary and Contributions: The paper introduces a generative model aimed at learning lower dimensional manifolds embedded in a larger space. They learn the mapping to the lower dimensional manifold using a usual auto-encoder with a reconstruction loss but with a specific architecture (M-flow): the decoder uses invertible normalising flow blocks, the encoder is the inverse of it, and the projection to lower dimensional space is done by zero-ing out a subset of the latent dimensions. They also introduce a variant (M_e-flow), where the encoder is not constrained to be the exact inverse of the decoder. Finally, to correctly estimate density in the lower dimensional manifold, they use another normalising flow on the latents. They argue that such a model can more faithfully represent the underlying lower dimensional manifold data. Contributions: - Estimating the density for their model, while tractible, is computationally expensive in higher dimensional spaces. They show with examples that likelihood maximisation for their model suffers from pathological cases where the learnt manifold can collapse, and introduce a simpler algorithm of separately learning the manifold and density on manifold can alleviate this. - They perform experiments to empirically compare it to normalising flows, as well as a pseudo-invertible variant, PIE on learning a polynomial surface, a real world particle physics distribution and high-dimensional images. ==== Post rebuttal and discussion update === Based on the rebuttal, I think the experimental evidence to show M-flow improves over the other baselines in the paper (PIE and AF) is sufficient, at the same time I'd strongly encourage the authors to do the simple auto-encoder/VAE baselines to more clearly understand if the invertibility of the decoder helps. I'm happy to have the paper be accepted, and I've updated my score from 5 to 6 to reflect this.

Strengths: The work shows a simple algorithm for learning lower dimensional manifolds, by combining a novel architecture that uses normalising flow blocks in auto-encoder learning. The theory portion is well grounded and correct. Their contribution is distinct from past work like PIE both architecturally, as well as in the algorithm used to train. They explore a range of toy tasks and metrics to evaluate the strengths and weaknesses of their model in comparison to flows and also PIE. In the polynomial task, they model the surface better than PIE. In stylegan experiments they obtain better samples than PIE, and more details in their samples than AF and also better FID scores.

Weaknesses: - A key architectural choice is the use of invertible flow blocks in the decoder of the auto-encoder. Just like they introduce M_e flows, what happens if they use a non-invertible decoder too ie just a simple auto-encoder? It would have been nice to see comparisons to this simple baseline auto-encoder in its ability to faithfully learn a lower dimensional manifold. Especially since they dont use the density estimation ability of flows in the auto-encoder itself (as their specific Jacobian [eqn 19 supplementary] is computationally challenging) - The experiments are slightly weak. In polynomial surface task, it seems the AF baseline also learnt the manifold? It has a low manifold distance, 0 reconstruction error and good AUC, it would’ve been nice to include its visual in Fig 6 Supplementary to confirm if it indeed learnt it. In particle physics task, both AF and PIE learnt a more accurate distribution in terms of sample quality / density estimation, so it seems unfair to instead compare on a new metric of inference of underlying parameters, since the overall aim of the paper seemed to be to better approximate low manifold distributions. Finally, in the real world task of CelebA, the AF performed better / as well as the M-flow (table 4), so its unclear if projecting to lower dimensional manifolds as suggested in the M-flow approach is indeed better than a fully invertible flow for high-dimensional real world distributions. - Lack of a metric to measure model performance: They show likelihood isnt a useful metric for their model (and infact bad to optimise directly), however it would be nice to have a different metric thats easy to evaluate and compare variants of their model.

Correctness: The theoretical analysis and experimental methodology are correct. The claims of improving upon autoencoders/AF could be more thoroughly proven.

Clarity: Paper is mostly well communicated. It does need frequent references to supplementary to fully understand the algorithms and experiments though.

Relation to Prior Work: In terms of experiments and explanation, the paper explains clearly how their approach is different from past work like AF/PIE. It would’ve been nice to see a more expanded related work that compares it to other approaches at modifying auto-encoders to learn lower dimensional manifolds, in particular VAE’s / VQ-VAE’s.

Reproducibility: Yes

Additional Feedback: Nits: - Connection to GANs isnt really strong, apart from the fact that the decoder in your auto-encoder maps a lower dimensional latent space to a bigger dimensional target space. - You claim denoising ability, but it isnt really shown, you only show reconstructions like other auto-encoders. You could add experiments the directly noise inputs and show that your model does a better job bringing it back to manifold. - Fig 10 Supplementary left side last row first image shows a boy with a green background in the M_e flow n=2 samples, but thats not really there in the manifold shown in the rightmost picture in Fig 9 Supplementary, are these different experiments?


Review 2

Summary and Contributions: The paper introduces a new method for modeling the lower dimensional manifold of data, and the density of the data on that manifold: Manifold flows (M-flows). The model is based on normalizing flows, but contrary to most normalizing flow methods, it is not optimized with a pure log-likelihood objective. In contrast, it is optimized with two optimization stages: 1. The manifold learning stage where a reconstruction error is measured between the data and its reconstruction after mapping it to the lower D manifold. 2. Optimizing the log-likelihood of the lower-dimensional mapping of the data. The authors argue with a clear example that one should be careful with the interpretation of the likelihood of the model and that optimizing only this as an objective can be problematic. They show that the method can be used instead as a pure generative model, or to do posterior inference when modeling conditional densities. The method is evaluated on several examples where the dimensionality of the manifold is known: toy examples, a particle physics dataset and a dataset generated by a StyleGAN. The method is also evaluated on CelebA, for which the dimensionality of the manifold is not known. ----------- Post-rebuttal update ------------- I think this paper is definitely above the acceptance threshold. As far as I am concerned the paper leaves some open questions/research directions, such as the influence of tuning the data dimensionality in M-flows and comparison with auto-encoders, but sufficiently compensates for this by containing evaluations of the method on a variety of tasks and by providing other contributions that I think can spark interest in the NeurIPS community. I therefore uphold my score for acceptance of this paper.

Strengths: 1. The proposed methods are evaluated on a variety of tasks. This helps convince the reader that realistic examples exist where the dimensionality of the manifold is known and that the proposed method has several applications. M-Flows are shown to perform well, especially when the manifold dimensionality is known. 2. The authors introduce a modification to a related method called PIE for the application of modeling conditional densities. This increases the strength of their baseline. 3. Figure 4 in the supplementary material is a very helpful concrete example that explains why optimizing the pure log likelihood will not lead to the model learning the correct manifold. 4. The paper is very well written with several pedagogical examples in the supplementary material.

Weaknesses: 1. Results in table 4 indicate that for datasets where the dimensionality of the manifold is unknown, regular flows (ambient flows) trained with log-likelihood produce more realistic images. As the authors state, this could be due to a suboptimal choice of the manifold dimension of M-flows. Although this is a weakness of the method, I believe it can also spark more research in this direction, as the method definitely seems promising for data with a known manifold dimensionality.

Correctness: As far as I can tell, the derivations are correct and the claims made in the paper are validated by the experiments. As a disclaimer: I do not have in-depth knowledge on the particle physics dataset/task.

Clarity: Yes, the paper is very well written. Many pedagogical toy examples are used to explain arguments made in the paper.

Relation to Prior Work: Yes.

Reproducibility: Yes

Additional Feedback: 1. In lines 245-248 the authors discuss a fair comparison between the different methods and mention their effort to keep the total number of coupling layers the same between several methods the same. Can the authors please also comment on the difference in the number of parameters? As the coupling layers in M-Flows don’t always act on data of the same dimensionality as regular AF flows, the number of parameters can be different, even with the same number of coupling layers. 2. Why are there no PIE results for CelebA? 3. For the celebA dataset, have you tried to train M-Flows with different n then 512? 4. Can you explain in the main text on a high level why including the SCANDAL loss consistently leads to a larger closure for all methods (lower closure is better). 5. In general, since the supplementary material contains so much more material, it would help the reader if you refer more frequently to the relevant parts of the supplementary material in the main text. For example, in the toy example in Figure 4 please refer to the appendix for the visual comparison to the learned densities of PIE in figure 6 of the supplemental material. The Gaussian circle and Lorenz attractor examples also seem not to be mentioned in the main text. Similarly, in the appendix different training strategies for learning the manifold and density are discussed. I don’t think this is mentioned in the main text. It would be nice to refer to your exploration efforts in this direction with one sentence in the main text, so that readers do know you have looked into this. 6. The use of both f and g in the paper text is a bit confusing at times. I understand the difference between them, but as an example of confusing notation: in the main text in lines 186-190 the weights of the manifold learning transformation are denoted with phi_f. In the supplementary material in section 3.A these parameters are referred to as phi_g. Sticking with either g or f in the main text unless it is really necessary to name the other would be helpful. 7. In table 3 the reconstruction error for AF is indicated with ‘-’ for the result without SCANDAL and ‘0’ for the result with SCANDAL. Please make this consistent. 8. Line 255: typo “metric” → “metrics”


Review 3

Summary and Contributions: The authors propose manifold-learning flows, which first learns a low dimensional. representation of data, and then performs tractible density estimation on this low dimensional space. Uniquely, they first compress down into a manifold space with a normalizing flow with zero padded dimensions. They then perform density estimation using another normalizing flow in this lower dimensional space.

Strengths: It seems to me that there are several advantages of this approach over a more naive setup such as autoencoding to obtain lower dimensions. For instance the NF based dimensionality reduction is invertible and has a 1-1 correspondence of points on the manifold to the latent space. Autoencoders don't define the points on the manifold separately from those off of it in a precise way. However, this is not shown well given that they only compare to flow baselines. They do show advantages in comparison to standard normalizing flows and a model “Pseudo Invertible Encoder”. To solve training difficulties, they alternate between training the manifold network and the distribution on top of this. This is an interesting concept with clear benefits when we know the data belongs to a low dimensional manifold. T

Weaknesses: I would want to see more comparisons against non-flow based methods such as autoencoders, cycleGANs etc. in learning density in lowered dimensions. After feedback: I am glad the authors agree and hope they will add such comparisons. Furthermore the authors somewhat dismiss the concern of estimating manifold density. Clearly in an n. dimensional space, any n-1 dimensional hyperplane (i.e., setting one coordinate to zero) would have zero measure, so it would have no volume at all and would not create a clear flow unless the intrinsic dimensionality maintained. I would actually want to see practically which estimation methods work best. After feedback: the authors still do not address this question, and I believe there are very few cases where the intrinsic dimensionality is known so I would want this added. The benefits as a general generative model (such as for images) is less clear to me without further experiments on computational efficiency and FID scores while training the full model not based on styleGAN latent spaces. Since a large advantage of this method is “tractability” it would be useful to compare the computation time of various methods.

Correctness: Yes

Clarity: The paper is fairly clearly written.

Relation to Prior Work: This can be improved by discussing dimensionality reduction and manifold learning literature in greater detail (diffusion maps, kernel PCA, autoencoders, etc).

Reproducibility: Yes

Additional Feedback: The paper should intuitively and clearly motivate using normalizing flows for dimensionality reduction as this is the most novel and most unintuitive part of the paper.


Review 4

Summary and Contributions: This paper propose a manifold-learning flows that simultaneously learn the data manifold as well as a tractable probability density on that manifold. They present a new training algorithm that separates manifold and density updates. The main attractive point in this work is they add an interesting reconstruction error (in supplementary (22)) to gradient flow problem.

Strengths: 1. This approach relaxes the requirement of knowing a closed-form expression for the chart from latent variables to the data manifold and instead learns the manifold from data. So M-flows may more accurately approximate the true data distribution, avoiding probability mass off the data manifold. 2. M-flows ensure that for data points on the manifold the encoder and decoder are the inverse of each other. 3.The projection onto the data manifold provides dimensionality reduction and denoising capabilities.

Weaknesses: 1. As discussed in appendix below eq.(23) :''An important choice is how these ........followed by the density-defining transformation h. ' ' My understanding is there are two schemes available to solve the optimisation: sequential and alternating. My main concern is I didn't found the sequential method comparison with alternating method in the experiment although the there are many experiments. Which one is better? sequential or alternating? 2. Adding the "reconstruction error" as a regularization could indeed learn the manifold structure of data. What is the convergence result of the proposed method by adding "reconstruction error"? Could you prove the empirical distribution with these particles convergences to the optimal distribution on manifold in terms of Wasserstein distance? I think this is a nontrivial problem but it is necessary. Conclusion: The experimental results seem promising however the theory for the approach is not clear. I think fixing some of the points mentioned above could greatly improve the clarity of the paper and make it a stronger submission. In the current state I don't think the paper is rigorous enough to be accepted.

Correctness: Yes

Clarity: Yes

Relation to Prior Work: yes

Reproducibility: Yes

Additional Feedback: I checked the response. The numerical part of this work is convicing. But the statistical properties (convergence to the true manifold or density) about this model should be further discussed and analysed. So my score is not changed.

[Author Response · NeurIPS 2020]

We thank the reviewers for their insightful feedback! We are encouraged that they appreciated the novelty of $\mathcal{M}$-flows (**R1**, **R3**, **R4**), their ability to learn the data manifold and a tractable density on it from samples (**R4**), and the exact invertibility on the manifold (**R3**, **R4**). We are glad that the reviewers liked our discussion of the subtleties of maximum-likelihood training (**R1**, **R2**) and appreciated the benefits of the new training scheme that separates manifold and density updates (**R1**, **R2**, **R3**, **R4**). They also found our experiments diverse (**R1**, **R2**) and convincing (**R2**, **R4**), noticed our improvements to the PIE baseline (**R2**) and commended the writing and pedagogical examples (**R2**). We answer some questions below, but will incorporate all feedback in the final version.

**Why use normalizing flows for dimensionality reduction (R3)?** Our primary goal was to construct a tractable probabilistic model for data on an $n$-dimensional manifold embedded in the data space, but in addition the coordinates of the learned manifold make a great candidate for dimensionality reduction: the flow approach ensures that for data exactly on the manifold, the compression to these variables is lossless, and the decoder is by construction the exact inverse of the encoder (unlike in autoencoders). Our goal is *not* to reduce the dimensionality further below $n$.

**What are the convergence properties of the proposed training method (R4)?** We wholeheartedly agree with **R4** that this needs more discussion. The convergence of the model to the correct manifold shape (defined by $f$) and to the true density on the manifold (defined by $f$ and $h$) can be analyzed separately. 1) The ability of $\mathcal{M}$-flows to converge to the correct *manifold* is essentially the same as the considerations for autoencoders, with an additional architectural requirement of invertibility. For data on a manifold that can be described by a single chart with the latent space dimensionality (which we assume in this first work), this does not pose a restriction (related to the fact that all submanifolds that satisfy modest regularity conditions can be expressed as level sets of bijections). 2) If $f$ has converged and learned the manifold, then learning the *density* on the manifold is a $n$-dimensional density estimation task. By implementing $h$ as a flow that is a universal density approximator, we ensure that the $\mathcal{M}$-flow model can express any density on the manifold (up to some regularity assumptions). We do not study the convergence properties in detail, but argue that the loss will learn the correct distribution in the infinite capacity, asymptotic training limit. In that spirit, the argument is much like the initial claims for the ability of GANs to learn the distribution on a data manifold.

**Is the sequential or alternating training scheme better (R4)?** Unclear. We compare both approaches in the polynomial surface experiment, but did not find a clear advantage.

**Comparison to autoencoders (AEs), VAEs, GANs (R1, R2, R3).** We agree that a comparison to (V)AEs and GANs on generative and dimensionality reduction tasks would be very interesting. We also really like **R1**'s suggestion of comparing to an $\mathcal{M}$-flow-like model with a non-invertible decoder. In the paper we focused on AF and PIE because (like $\mathcal{M}$-flows) they allow for exact likelihood evaluations, which is crucial for inference tasks.

**It would be nice to have a different metric to compare the models (R1).** We agree, but do not know any single metric of all relevant aspects. Since likelihood values on different manifolds cannot be compared, we chose to study different metrics of data generation (manifold distance, FID scores, physics closure tests), manifold quality (reconstruction error when projecting to the manifold), inference (MMD between true and estimated posterior, log posterior evaluated at the true parameter point), and OOD detection (ROC AUC between in-distribution and OOD test samples).

**In the particle physics task, it seems unfair to compare on a new metric of inference of underlying parameters (R1).** Instead of "unfair" we would characterize it as a different metric that is often more relevant in a scientific context. *Likelihood-free inference* (LFI) is its own thriving research area with applications from neuroscience to epidemiology. Many state-of-the-art methods do not involve learning the likelihood, so the quality of likelihood estimation is not admissable to compare them. The good performance of $\mathcal{M}$-flows on LFI tasks could be impactful.

**Did the AF baseline learn the manifold in the polynomial surface task (R1)?** No. On the right we show the AF and $\mathcal{M}$-flow log likelihood along the two-dimensional slice $x_0 = 0$ through the 3-D data space. The AF density is sharply peaked around the true data manifold and most of the probability mass is very close to it. Still, it has non-zero support off the manifold, especially in regions of low density. In contrast, the $\mathcal{M}$-flow exactly learns a two-dimensional manifold. We thank **R1** for the suggestion of showing this explicitly and will include more results in the final version.

**No PIE results for CelebA (R2).** These were not completed in time for the submission. We have the answers now: on CelebA, PIE achieves FID scores of $75.7 \pm 5.1$, substantially worse than our $\mathcal{M}$-flow results and the AF baseline.

**$\mathcal{M}$-flows did not outperform AF on CelebA (R1, R2).** Yes. As **R2** pointed out, here we do not know the manifold dimension. Due to limited resources, we have not scanned over this hyperparameter or optimized the architecture. The good performance of $\mathcal{M}$-flows in datasets with known manifold dimension makes us optimistic that such a tuning will improve the results, but we are not in a position to make this claim at this time. We share **R2**'s hope that our results will spark more research along these lines, and were excited to see some steps at the recent ICML INNF+ workshop.

[Meta-Review · NeurIPS 2020]

All reviewers agree that the presented technique for simultaneous manifold and density estimation is interesting and novel. However, they also agree that the paper leaves important questions open. While one of the reviewers would like to see a stronger statistical analysis before acceptance, the others believe that the paper is above acceptance threshold and that the community would benefit from its communication. The meta-reviewer agrees. To address the concerns of the reviewers, the camera-ready paper needed to include at least the following results: 1. Include results that investigate if the invertible nature of the normalising flow in the decoder is useful by e.g considering a version of the Me-flow where g is not constrained to be invertible. In the same vein, a comparison with a simple VAE baseline should be included. 2. Include results that demonstrate how the latent dimension n can be chosen by cross-validation. Investigate how the results on CelebA depend on the latent dimension n.